# LINK: Learning Instance-level Knowledge from Vision-Language Models for Human-Object Interaction Detection

**Eastman Z Y, Wu**[1,2,3], **Yali Li**[1,2,3], **Yuan Wang**[1,2,3], **Shengjin Wang**[1,2,3]*

[1] Department of Electronic Engineering, Tsinghua University
[2] Beijing National Research Center for Information Science and Technology (BNRist), China
[3] National Engineering Research Center of Dangerous Articles and Explosives Detection
Technologies, Beijing 100084, China
{wu-zy23, wy23}@mails.tsinghua.edu.cn
{liyali13, wgsgj}@tsinghua.edu.cn

## Abstract

Human-Object Interaction (HOI) detection with vision-language models (VLMs) has progressed rapidly, yet a trade-off persists between specialization and generalization. Two major challenges remain: (1) the sparsity of supervision, which hampers effective transfer of foundation models to HOI tasks, and (2) the absence of a generalizable architecture that can excel in both fully supervised and zero-shot scenarios. To address these issues, we propose **LINK**, **L**earning **IN**stance-level **K**nowledge from VLMs. First, we introduce a HOI detection framework equipped with a Human-Object Geometrical Encoder and a VLM Linking Decoder. By decoupling from detector-specific features, our design ensures plug-and-play compatibility with arbitrary object detectors and consistent adaptability across diverse settings. Building on this foundation, we develop a Progressive Learning Strategy under a teacher-student paradigm, which delivers dense supervision over all potential human-object pairs. By contrasting subtle spatial and semantic differences between positive and negative instances, the model learns robust and transferable HOI representations. LINK sets new state-of-the-art on SWiG-HOI, HICO-DET, and V-COCO across zero-shot, fully supervised, and open-vocabulary settings, with strong generalization to unseen interaction categories.

## 1 Introduction

Human-Object Interaction (HOI) detection has recently emerged as a rapidly developing field, requiring higher-level visual understanding beyond standard object detection. By focusing on complex human-centric interactions, HOI detection is essential for applications such as intelligent robotics and anomalous behavior detection (Liu et al., 2018). Its core goal is to localize human-object pairs and recognize their interactions as structured triplets: $< human, action, object >$.

Recently, foundation models pretrained on large-scale multimodal datasets have shown strong capabilities to provide effective feature representations for downstream tasks. In the field of HOI detection, numerous studies (Lei et al., 2023; Cao et al., 2024; Ning et al., 2023; Mao et al., 2023; Lei et al., 2025b; Wu et al., 2023) have successfully used pretrained CLIP models (Radford et al., 2021) to improve the recognition of rare and unseen interactions, thereby advancing zero-shot and few-shot learning in HOI detection.

However, existing VLM-based HOI detectors often face an inherent trade-off between *specialization* and *generalization*. Dedicated architectures are typically optimized for fully supervised benchmarks, yielding strong in-domain performance but struggling in zero-shot and cross-domain settings due to limited generalization capacity. Conversely, zero-shotoriented methods are commonly built upon CLIP with lightweight modifications. While effective in recognizing novel HOI categories,

---

*Corresponding author.

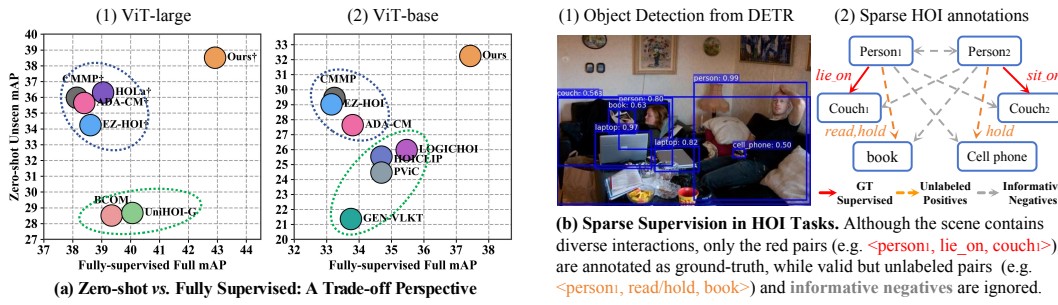

Figure 1: Illustration of challenges in HOI detection.

their limited task-specific adaptability leads to suboptimal performance under fully supervised evaluation. As illustrated in Figure 1(a), a clear trade-off emerges: methods achieving higher fully supervised performance often suffer a decline in their corresponding zero-shot performance. A second challenge lies in the sparsity of supervision, as shown in Figure 1(b). In visual scenes, humans and objects form densely connected graph structures, yet ground-truth annotations cover only a small subset of edges, leaving most instances under-utilized. These ignored cases include not only valid but unlabeled pairs, but also informative negatives that could contribute to more robust learning. Together, these issues highlight the core difficulty of adapting VLMs, pretrained on large-scale imagetext pairs, to instance-level HOI detection under sparse supervision.

To address these challenges, we propose **LINK**, **L**earning **IN**stance-level **K**nowledge, which integrates architectural innovations with a progressive learning strategy. First, we design a HOI detector that introduces a *Human-Object Geometrical Encoder* to capture the spatial relationships of paired human-object instances, and a *VLM Linking Decoder* that bridges VLMs with HOI detection by transforming global semantic representations into fine-grained, instance-level HOI patterns. Second, we develop a *Progressive Learning Strategy*. In the first stage, the model is trained with standard supervision to adapt intrinsic knowledge within VLM into HOI-specific patterns. In the second stage, we leverage this pre-trained model as the teacher, where the student receives sparse GT supervision while all human-object pairs, including negatives, are further guided by dense distillation losses. By contrasting subtle spatial and semantic differences between positive and negative instances, the model learns to resolve ambiguities and acquire more discriminative HOI representations.

Our **LINK**, offers key advantages. It maintains consistency across diverse scenarios (fully supervised, zero-shot, and open-vocabulary) and captures fine-grained spatial and semantic patterns for robust HOI prediction. Even in cross-domain transfer to synthetic images with drastic semantic shifts, preserved spatial patterns support reliable decisions. In summary, our main contributions are:

- We propose a HOI detector with a *Human-Object Geometrical Encoder* and a *VLM Linking Decoder*. This design strengthens HOI-specific reasoning capacity while avoiding unnecessary complexity that may compromise generalization. By decoupling from detector-specific features, **LINK** achieves plug-and-play compatibility with arbitrary object detectors without fine-tuning.

- We introduce a progressive learning strategy that delivers dense supervision to all candidate human-object pairs, enabling the model to capture fine-grained spatial and semantic distinctions between positive and negative instances. This effectively mitigates the supervision sparsity inherent in HOI tasks.

- We conduct the first comprehensive evaluation of HOI detection across diverse foundation models (CLIP, BLIP, DINOv2, DINO@448, SigLIP2, Florence2), and demonstrate that **+LINK** consistently improves all baselines, with the most substantial gains on long-tail HOIs ($\leq 10$ samples).

Our method outperforms existing methods by a large margin. For instance, on HICO-DET with an R50 backbone and ViT-L CLIP, LINK achieves **42.92 / 45.03** mAP on the full / rare sets, surpassing the previous state-of-the-art by **3.87 / 6.37** mAP, corresponding to relative gains of **9.9% / 16.5%**. Moreover, when scaled up to a Swin-L backbone, LINK further improves to **49.06 / 53.63** mAP.

## 2 RELATED WORKS

**Human-Object Interaction Detection:** Human-Object Interaction (HOI) detection is a composite task that involves localizing humans and objects, as well as recognizing their interactions. Existing methods can be broadly classified into one-stage and two-stage paradigms. One-stage methods (Liao et al., 2020; Chen & Yanai, 2021; Chen et al., 2021; Kim et al., 2021; Zou et al., 2021) aim to jointly localize objects and infer interactions in a single forward pass. Early approaches, such as PPDM (Liao et al., 2020) and UnionDet (Kim et al., 2020a), leverage interaction points or union regions as anchors to guide localization and feature extraction. More recently, Transformer-based architectures have advanced the field by introducing query-based HOI detectors. In contrast, two-stage methods (Chao et al., 2018; Gao et al., 2020; 2018; Gkioxari et al., 2018; Gupta et al., 2019; Kim et al., 2020b; Zhou & Chi, 2019; Liu et al., 2020; Wu et al., 2024) decouple the process into object detection followed by HOI classification for each human-object pair. This separation offers greater flexibility, interpretability, and modularity, and has gained increasing attention in recent work. Given the modular nature of the two-stage paradigm, it is particularly well-suited for designing generalizable and scalable HOI detectors. In this paper, we aim to develop a unified two-stage HOI detector that performs effectively across both specific benchmarks and generalization scenarios.

**Adapting Vision-Language Models:** The rapid advancement of vision-language models (VLMs) (Radford et al., 2021; Li et al., 2023a; 2022; Wang et al., 2021b; Zhang et al., 2022b) has recently demonstrated strong zero-shot capabilities. This has prompted growing interest in adapting VLMs for Human-Object Interaction (HOI) detection. For instance, HOICLIP (Ning et al., 2023) employs a query-based approach to harness visual knowledge from CLIP, achieving zero-shot HOI enhancement by leveraging CLIPs image-text retrieval capabilities. BCOM (Wang et al., 2024) proposes an occlusion-aware Contextual Mining method that guides the model to recover spatial details from occluded feature maps, thereby improving robustness in crowded scenes. ADA-CM employs a Concept-guided Memory to retrieve both domain-specific and domain-agnostic knowledge from CLIP, enabling a quick adaptation to datasets. More recently, CMMP (Lei et al., 2025b) introduces conditional multi-modal prompts enriched with priors to decouple visual representation and interaction classification, thereby enhancing the zero-shot HOI detection capability of CLIP-based models.

## 3 METHOD

In this section, we first review the problem formulation in VLM-based two-stage HOI detection methods (3.1). We then introduce our proposed unified HOI architecture (3.2), which consists of a Human-Object Geometrical Encoder and a VLM Linking Decoder. Next, we present our Progressive Learning Strategy (3.3).

### 3.1 PRELIMINARY: PROBLEM FORMULATION.

The architecture of a two-stage VLM-based HOI detection framework is illustrated in Figure 2. In this paradigm, an off-the-shelf object detector is first employed to localize entities. Following standard practice, DETR is used to generate all bounding boxes $\mathcal{B}$, which are divided into human boxes $\mathcal{B}_h$ and object boxes $\mathcal{B}_o$. For each box, a corresponding featureeither object queries in DETR or ROI-aligned featuresis extracted and used as a unary query for subsequent interaction reasoning. This results in the mappings $Q_h \leftrightarrow \mathcal{B}_h$ and $Q_o \leftrightarrow \mathcal{B}_o$. By enumerating all possible human-object pairs, we construct $Q_{h\text{-}o}$, where each query corresponds to a pair of boxes $[\mathcal{B}_h, \mathcal{B}_o]$.

We then refine $Q_{h\text{-}o}$ via an encoder-decoder architecture as $Q_{h\text{-}o} = \text{Decoder}(\text{Encoder}(Q_{h\text{-}o}), \mathcal{F})$, where $\mathcal{F}$ denotes external features, such as feature maps extracted from CLIP or backbone. The output interaction logits are predicted by $L_{h\text{-}o} = \text{FFN}(Q_{h\text{-}o})$. As a result, for each human-object pair, the model outputs a prediction in the form of $\langle \mathcal{B}_h, L_{h\text{-}o}, \mathcal{B}_o, C_o \rangle$, where $C_o$ is the object category. In the practical training process, supervision is applied only to valid queries $Q^m$those where both the human and object bounding boxes have an Intersection over Union (IoU) above 0.5 with ground-truth boxes. Formally, matched queries are denoted as $\mathcal{M}$ and defined as:

$$\mathcal{M} = \left\{ Q_{h\text{-}o} \, \middle| \, \text{IoU}(\mathcal{B}_h, \mathcal{B}_h^{\text{gt}}) \geq 0.5 \ \wedge \ \text{IoU}(\mathcal{B}_o, \mathcal{B}_o^{\text{gt}}) \geq 0.5 \right\} \tag{1}$$

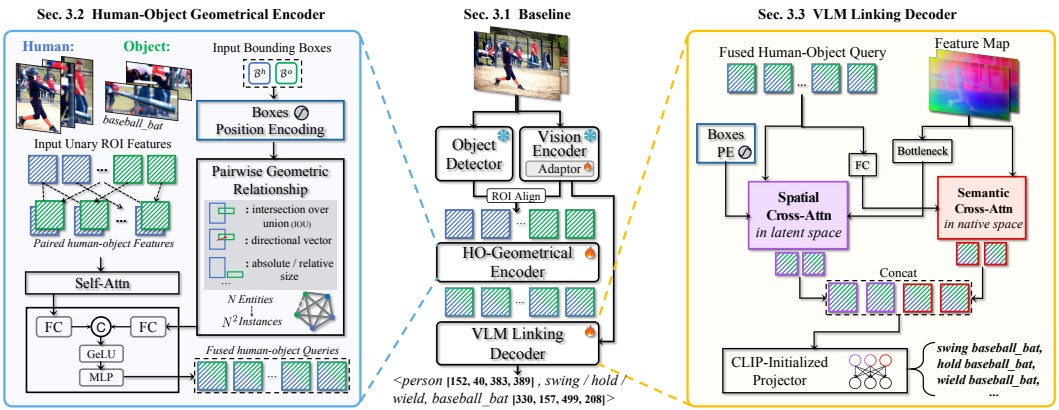

Figure 2: Overview of the proposed LINK framework. **(Sec. 3.1)** Baseline: a VLM-based two-stage HOI detection pipeline. **(Sec. 3.2)** Human-Object Geometrical Encoder: integrates ROI features, bounding-box encodings, and pairwise geometry to encode human-object spatial dependencies. **(Sec. 3.3)** VLM Linking Decoder: refines pairwise queries via geometry-aware and semantic cross-attention, and predicts HOI triplets with a CLIP-initialized head.

Where $\mathcal{B}_h$, $\mathcal{B}_o$ are predicted human and object boxes, respectively. The training objective is to optimize the parameters $\theta$ by minimizing the expected HOI classification loss over matched queries:

$$\theta^* = \arg\min_\theta \mathbb{E}_{\mathcal{I} \sim \mathcal{X}} \left[ \mathcal{L}_\mathcal{M}(\Phi_\theta(\mathcal{I}, \mathcal{B}), \mathcal{GT}) \right], \tag{2}$$

where $\Phi_\theta$ denotes the HOI detector, $\mathcal{L}_\mathcal{M}$ is the multi-label classification loss computed over the matched queries $\mathcal{M}$, and $\mathcal{X}$ is the training dataset, $\mathcal{I}$ represents the input image.

## 3.2 MODEL ARCHITECTURE.

**Object Detector and Vision Encoder.** To ensure architectural generality, we do not use the dataset-specific object queries produced by a fine-tuned DETR. Instead, we obtain unary queries via ROI Align, using the feature maps extracted from a VLM pretrained on large-scale data and the detected bounding boxes. Specifically, given human and object boxes $\mathcal{B}_h$ and $\mathcal{B}_o$, we apply ROI Align on the feature map $F \in \mathbb{R}^{H \times W \times C}$ to extract the corresponding unary queries $Q_h$ and $Q_o$, both of dimensionality $C$.

**Human-Object Geometrical Encoder.** Since VLMs such as CLIP are pre-trained with image-level contrastive objectives, they primarily capture global semantic information. Compared to the object queries in DETR, their spatial awareness and region-level discrimination are relatively limited, which motivates the introduction of Geometry-aware query refinement. To this end, we encode each bounding box using its normalized center and size with sinusoidal positional encoding. Specifically, given a bounding box $\mathcal{B} = (x_1, y_1, x_2, y_2)$ and image size $(W, H)$, we normalize the box as $\hat{\mathcal{B}}$, We then compute the center and size as $\mathcal{C}$ and $\mathcal{S}$: $\hat{\mathcal{B}} = \left( \frac{x_1}{W}, \frac{y_1}{H}, \frac{x_2}{W}, \frac{y_2}{H} \right)$, $\mathcal{C} = \frac{1}{2}(\hat{x}_1 + \hat{x}_2, \hat{y}_1 + \hat{y}_2)$, $\mathcal{S} = (\hat{x}_2 - \hat{x}_1, \hat{y}_2 - \hat{y}_1)$. Apply sinusoidal encoding to each: $\text{PE}(\mathcal{B}) = \text{PE}(\mathcal{C}) \oplus \text{PE}(\mathcal{S})$, where $\text{PE}(\cdot)$ is the standard 2D sinusoidal positional encoding function and $\oplus$ denotes vector concatenation. The resulting spatial embeddings are added to the unary queries $Q_h$ and $Q_o$, and further refined through a self-attention mechanism. This process can be expressed as: $Q = \textit{Self-Attn}(Q + \text{PE}(\mathcal{B}))$, Next, we construct pairwise human-object queries $Q_{h\text{-}o}$. Specifically, we iterate over all possible human-object combinations and concatenate their features to form paired queries, expressed as:

$$Q_{h\text{-}o} = \text{Linear}(\text{C}[Q_i, Q_j]), \text{where } i \in H, j \in O \cup H. \tag{3}$$

Here, $H$ and $O$ denote the sets of detected humans and objects, respectively. The notation C denotes concatenation. The inclusion of $j \in H$ allows the model to capture human-human interactions.

Previously, we defined unary positional encodings. To enrich the pairwise representation with instance-level spatial awareness, we further encode the geometric relation between each human-object pair using pairwise spatial encoding, following UPT (Zhang et al., 2022a). This produces a

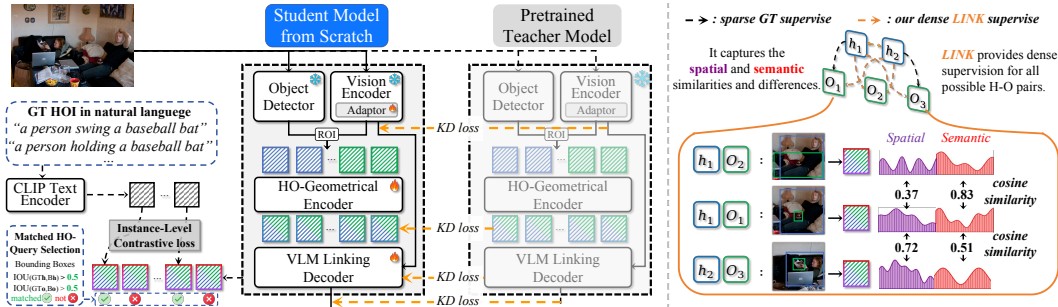

Figure 3: **Progressive Learning Strategy** (a) Student-teacher framework: the student model is trained from scratch with sparse GT supervision, while the pretrained teacher provides dense guidance via multi-level knowledge distillation. (b) Three levels of knowledge alignment: logits-level, query-level, and feature-map level, where instance-level matching enables fine-grained supervision on spatial and semantic representations.

spatial relation vector $R_{i,j}$ for each pair $(i, j)$. We then fuse the queries $Q_{h\text{-}o}$ with their corresponding spatial encodings $R_{i,j}$ through a Multi-Modal Fusion (MMF) module. The fusion process is defined as: $x = \text{LN}_1(\text{FC}_1(Q_{h\text{-}o}))$, $y = \text{LN}_2(\text{FC}_2(R_{i,j}))$ and $z = \text{MLP}(\text{ReLU}(\text{Concat}[x, y]))$, where $\text{FC}_1, \text{FC}_2$ are linear projections to a shared embedding space, $\text{LN}_1, \text{LN}_2$ are LayerNorm layers, and MLP is a multi-layer perceptron that produces the final fused representation. The output $z$ serves as the final output of the encoder, i.e., the refined pairwise query $Q_{h\text{-}o}$.

At this stage, we decouple the features from the object detector and derive queries solely based on VLM features and detected bounding boxes, thereby mitigating detector-induced biases. Meanwhile, it enhances spatial awareness and produces paired H-O instances for subsequent reasoning.

**VLM Linking Decoder.** In the decoder, we perform cross-attention between the pairwise queries $Q_{h\text{-}o}$ and the VLM feature map $F \in \mathbb{R}^{H \times W \times C}$ to aggregate both spatial cues and semantic information. Building on standard cross-attention, we design a VLM Linking structure consisting of a *spatial branch* and a *semantic branch*.

The **spatial branch** reduces the dimensionality of the feature map through a connector module, forming a latent bottleneck that interacts with $Q_{h\text{-}o}$. This branch focuses on capturing fine-grained geometric relationships and spatial cues within the scene. To further enhance spatial reasoning, we adopt the attention-guidance mechanism proposed in PViC (Zhang et al., 2023), where positional encodings derived from bounding boxes are used to constrain the attention maps.

In contrast, the **semantic branch** expands $Q_{h\text{-}o}$ and performs attention in the high-dimensional native space of the VLM, enabling richer aggregation of high-level semantics. While the spatial branch enhances fine-grained modeling, the semantic branch leverages global representations to improve transferability. The outputs of both branches are fused by concatenation followed by a feed-forward network:

$$Q_{h\text{-}o}^n = \text{Linear}(Q_{h\text{-}o}), \quad F^l = \text{MLP}(F), \tag{4}$$

$$Q^{\text{out}} = \text{MLP}\left(\text{CA}^{\text{be}}(Q_{h\text{-}o}, F^l) \copyright \text{CA}(Q_{h\text{-}o}^n, F)\right), \tag{5}$$

where $F^l$ denotes the compressed feature map from the latent branch, MLP is a multi-layer perceptron used for final fusion, and $\copyright$ indicates concatenation. Here, $\text{CA}(\cdot)$ refers to the standard cross-attention, while $\text{CA}^{\text{be}}(\cdot)$ is a box-encoding-guided modified cross-attention. Finally, $Q^{\text{out}}$ is passed through an FFN initialized with CLIP text embeddings to generate the final HOI logits.

## 3.3 PROGRESSIVE LEARNING STRATEGY

We propose a unified architecture that establishes a strong baseline for adapting foundation models to HOI detection. Building upon this foundation, we further introduce a knowledge learning strategy based on a teacher-student paradigm to alleviate the challenge of sparse supervision.

First, we leverage our architecture to construct a pure baseline by pre-training it on HOI data using only the original ground-truth annotations, which serves as the **Teacher Model**. Thanks to the architectural design, the teacher model transforms the frozen image-level representations from VLMs into learnable instance-level representations tailored for HOI tasks.

Second, we employ the pre-trained model to perform teacher-student transfer paradigm, as illustrated in Figure 3 Left. **The Student Model is jointly supervised by both ground-truth annotations and guidance from the pre-trained teacher.** Since the teacher and student share the same input and architecture, we achieve one-to-one aligned human-object instances. This alignment enables knowledge transfer across all candidate human-object pairs, rather than being restricted to the limited subset defined by matched queries.

As shown in Figure 3, traditional supervision (black dashed arrow) covers only a limited set of annotated human-object instances. In contrast, our learning strategy provides adaptive supervision for all potential instances (orange dashed arrow), delivering richer and more comprehensive guidance. Moreover, it facilitates knowledge transfer across multiple levels, further enhancing the models ability to capture complex interaction patterns. We adopt knowledge transfer losses with Kullback-Leibler (KL) divergence: $\mathcal{KD}_{\text{KL}}(f_{\text{stu}}, f_{\text{t}}) = \text{KL}\left(\sigma(f_{\text{t}}/\tau) \,\|\, \sigma(f_{\text{stu}}/\tau)\right)$, where $f_{\text{stu}}$ and $f_{\text{t}}$ denote the student and teacher features or logits, $\sigma(\cdot)$ is the softmax function, and $\tau$ is a temperature factor.

**Feature Map.** We perform knowledge transfer at the feature map level by aligning the representations produced by the VLMs using a lightweight adapter. Let $F_{\text{stu}}, F_{\text{t}} \in \mathbb{R}^{C \times H \times W}$ denote the feature maps from the student and teacher models, respectively. To address potential mismatches in spatial and channel dimensions, we align the teachers feature map to match the students. Specifically, we first apply bilinear interpolation to adjust the spatial resolution: $F'_{\text{t}} = \text{Interpolate}(F_{\text{t}}, \text{size} = (H_{\text{stu}}, W_{\text{stu}}))$, followed by trilinear interpolation to align the channel dimension: $F''_{\text{t}} = \text{Interpolate}(F'_{\text{t}}, \text{size} = (C_{\text{stu}}, H_{\text{stu}}, W_{\text{stu}}))$. Both feature maps are then flattened into shape $HW \times C$, and the loss is computed as: $\mathcal{L}^{\text{feat}}_{\text{KD}} = \mathcal{KD}(F_{\text{stu}}, F''_{\text{t}})$.

**Query.** Thanks to our unified architecture, we design a fully mirrored student-teacher paradigm in which both the HO Geometrical encoder and VLM Linking decoder share identical structures. This symmetry enables query-level knowledge transfer across all human-object pair queries throughout the entire model. Formally, the query-level transfer loss is defined as:

$$\mathcal{L}^{\text{query}}_{\text{KD}} = \frac{1}{L_e} \sum_{\ell=1}^{L_e} \mathcal{KD}(\mathcal{Q}^{(\ell)}_{e,\text{stu}}, \mathcal{Q}^{(\ell)}_{e,\text{t}}) + \frac{1}{L_d} \sum_{\ell=1}^{L_d} \mathcal{KD}(\mathcal{Q}^{(\ell)}_{d,\text{stu}}, \mathcal{Q}^{(\ell)}_{d,\text{t}}), \tag{6}$$

where $L_e$ and $L_d$ denote the number of layers in the encoder and decoder, respectively. $\mathcal{Q}^{(\ell)}_{e,\text{stu}}$ and $\mathcal{Q}^{(\ell)}_{e,\text{t}}$ represent the sets of student and teacher queries at the $\ell$-th layer of the encoder. Similarly, $\mathcal{Q}^{(\ell)}_{d,\text{stu}}$ and $\mathcal{Q}^{(\ell)}_{d,\text{t}}$ denote the student and teacher queries at the $\ell$-th decoder layer. The function $\mathcal{KD}(\cdot, \cdot)$ computes the average token-wise distillation loss between corresponding query sets.

**Logits.** We perform knowledge transfer at the logits level, which provides multi-label interaction guidance for each human-object pair. This also captures richer contextual information in the distribution of logits over potential human-object interactions. Additionally, we follow previous two-stage methods (Zhang et al., 2023) and combine the predicted logits with detection confidence scores. The final score is computed as: $\Psi_s = \log\left(\frac{\mathscr{P}}{1+\exp(-O_s)-\mathscr{P}}\right)$, where $\mathscr{P}$ denotes the confidence score of the paired bounding boxes and $O_s$ is the predicted HOI logit. The distillation loss at the logits level is then defined as: $\mathcal{L}^{\text{logits}}_{\text{KD}} = \text{KD}\left(\Psi(F_{\text{stu}}), \Psi(F_{\text{t}})\right)$.

Now, we extend the original training objective by applying knowledge transfer across all human-object pairs and at multiple levels $\mathcal{G}$, as in the following objective:

$$\theta^* = \arg\min_{\theta} \mathbb{E}_{\mathcal{I} \sim \mathcal{X}} \left[ \mathcal{L}_{\mathcal{M}}(\Phi_\theta(\mathcal{I}, \mathcal{B}), \mathcal{GT}) + \sum_{g \in \mathcal{G}} \mathcal{KD}_g(\Phi_\theta(\mathcal{I}, \mathcal{B}), \Phi_{\text{t}}(\mathcal{I}, \mathcal{B})) \right] \tag{7}$$

where $\Phi_{\text{t}}$ is the teacher model and $\mathcal{KD}_g$ denotes the knowledge distillation loss at level $g$, which includes *feature map*, *query*, and *logits*-level supervision.

Table 1: **Zero-shot** performance comparison under RF-UC, NF-UC, and UO settings. HM denotes the harmonic mean. The best result in each column is in **bold**, the second best is underlined. The symbol † indicates results with CLIP-ViT-L as the VLM.

| Method | RF-UC | | | | NF-UC | | | | UO | | | |
|---|---|---|---|---|---|---|---|---|---|---|---|---|
| | HM | Unseen | Seen | Full | HM | Unseen | Seen | Full | HM | Unseen | Seen | Full |
| *Fully-supervised methods* | | | | | | | | | | | | |
| PViC (Zhang et al.) | 27.85 | 24.45 | 32.36 | 30.78 | 26.80 | 24.74 | 29.23 | 28.07 | 25.07 | 19.13 | 36.37 | 33.50 |
| HOICLIP (Ning et al.) | 29.40 | 25.53 | 34.85 | 32.99 | 27.22 | 26.39 | 28.10 | 27.75 | 21.28 | 16.20 | 30.99 | 28.53 |
| LOGICHOI (Li et al.) | 29.79 | 25.97 | 34.93 | 33.17 | 27.34 | 26.84 | 27.86 | 27.95 | 20.68 | 15.67 | 30.42 | 28.23 |
| GEN-VLKT (Liao et al.) | 25.91 | 21.36 | 32.91 | 30.56 | 24.19 | 25.05 | 23.38 | 23.71 | 15.42 | 10.51 | 28.92 | 25.63 |
| *Zero-shot oriented methods* | | | | | | | | | | | | |
| ADA-CM (Lei et al.) | 30.63 | 27.63 | 34.35 | 33.01 | 31.76 | 32.41 | 31.13 | 31.39 | – | – | – | – |
| CLIP4HOI (Mao et al.) | 31.59 | 28.47 | 35.48 | 34.08 | 29.72 | 31.33 | 28.26 | 28.90 | 32.25 | 31.79 | 32.73 | 32.58 |
| BCOM† (Wang et al.) | 31.45 | 28.52 | 35.04 | 33.74 | 32.43 | 33.12 | 31.76 | 32.03 | – | – | – | – |
| CMMP (Lei et al.) | 31.07 | 29.45 | 32.87 | 32.18 | 30.85 | 32.09 | 29.71 | 30.18 | 32.40 | 33.76 | 31.15 | 31.59 |
| EZ-HOI (Lei et al.) | 31.38 | 29.02 | 34.15 | 33.13 | 32.03 | 33.66 | 30.55 | 31.17 | 32.66 | 33.28 | 32.06 | 32.27 |
| HOLa (Lei et al.) | 32.69 | 30.61 | 35.08 | 34.19 | 33.35 | **35.25** | 31.64 | 32.36 | 34.65 | 36.45 | 33.02 | 33.59 |
| **LINK** | 33.42 | 32.25 | 34.68 | 34.19 | 34.07 | 33.72 | 34.42 | 34.25 | 33.73 | 34.05 | 33.41 | 33.66 |
| **LINK†** | **39.40** | **38.51** | **40.33** | **39.97** | **35.14** | 34.63 | **35.67** | **35.43** | **37.92** | **38.24** | **37.61** | **37.91** |

Table 2: **Zero-shot** comparison under UV setting.

| Method | UV | | | |
|---|---|---|---|---|
| | HM | Unseen | Seen | Full |
| *Fully-supervised methods* | | | | |
| PViC (Zhang et al.) | 23.84 | 19.58 | 30.48 | 28.95 |
| HOICLIP (Ning et al.) | 27.69 | 24.30 | 32.19 | 31.09 |
| LOGICHOI (Li et al.) | – | – | – | – |
| GEN-VLKT (Liao et al.) | 24.76 | 20.96 | 30.23 | 28.74 |
| *Zero-shot oriented methods* | | | | |
| ADA-CM (Lei et al.) | – | – | – | – |
| CLIP4HOI (Mao et al.) | 28.35 | 26.02 | 31.14 | 30.42 |
| BCOM† (Wang et al.) | – | – | – | – |
| CMMP (Lei et al.) | 29.13 | 26.23 | 32.75 | 31.84 |
| EZ-HOI (Lei et al.) | 28.64 | 25.10 | 33.49 | 32.32 |
| HOLa (Lei et al.) | 31.09 | 27.91 | 35.09 | 34.09 |
| **LINK** | 29.29 | 27.01 | 32.00 | 31.30 |
| **LINK†** | **31.92** | 27.22 | **38.36** | **36.88** |

Table 3: **Few-shot comparison** on HICO-DET and V-COCO datasets.

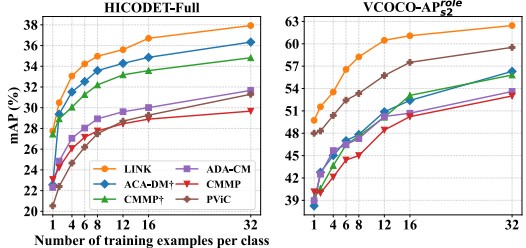

## 4 EXPERIMENTS

**Implementation Details.** We trained two scales of object detectors for fair comparison with prior work: DETR (Carion et al., 2020) with a ResNet-50 (He et al., 2016) backbone, and H-Deformable-DETR (Jia et al., 2023) with a Swin-Transformer-Large (Liu et al., 2021) backbone. Both detectors are pretrained on MS-COCO (Lin et al., 2014) and fine-tuned on the target datasets following standard practice. ROI-aligned features are extracted at a resolution of $7 \times 7$ using average pooling. We adopt AdamW with a weight decay of $10^{-4}$, training for 15 epochs with FocalBCE loss. The learning rate is decayed to 20% of its initial value after the $10^{\text{th}}$ epoch. For $\mathcal{KD}$ losses, the temperature coefficient is set to $\tau = 2.0$. In our main comparisons, we follow prior work and use CLIP as the visionlanguage backbone. The connector module projects features into a 384-dimensional space. Unless otherwise specified, the teacher model is CLIP ViT-L/14@336px. Additional details are provided in the supplementary material.

### 4.1 COMPARISON WITH STATE-OF-THE-ARTS

**Zero-shot Results.** Table 1 and 2 reports zero-shot performance under four standard settings: RF-UC (Rare-First Unseen Combination), NF-UC (Non-Rare First Unseen Combination), UO (Unseen Object) and UV (unseen Verb). HM denotes the harmonic mean between seen and unseen categories. Our method consistently outperforms both fully-supervised and zero-shot-oriented approaches. In our experiments, LINK uses CLIP ViT-Base for both teacher and student, while LINK† uses CLIP ViT-Large. Across the four zero-shot settings, LINK achieves two best and two second-best results among SOTA methods. Under the RF-UC setting, our method achieves an unseen score of **32.25**, surpassing the previous best result of 30.61 by +1.64. When scaling up to the ViT-Large, the performance further improves. Similarly, under the NF-UC and UO settings, our model attains harmonic means of **35.14** and **37.92**, respectively, establishing new state-of-the-art performance. These results

Table 4: **Comparison with state-of-the-art on HICO-DET and V-COCO.** Results are grouped into fully-supervised *v.s* zero-shot oriented methods. **Bold** indicates the best performance. All experiments are conducted under a fair setting where the teacher and student use the same scale VLM, such as CLIP-ViT-B to CLIP-ViT-B.

| Method | Configuration | HICO-DET (Default) | | | (Known Obj.) | | | V-COCO |
|---|---|---|---|---|---|---|---|---|
| | Backbone / VLM | Full | Rare | N-rare | Full | Rare | N-rare | $AP_{role}^{S2}$ |
| *Fully-supervised methods* | | | | | | | | |
| KI2HOI (Xue et al.) | R50 / CLIP-B | 34.20 | 32.26 | 36.10 | 37.85 | 35.89 | 38.78 | 65.0 |
| HOICLIP (Ning et al.) | R50 / CLIP-B | 34.69 | 31.12 | 35.74 | 37.61 | 34.47 | 38.54 | 64.8 |
| CLIP4HOI (Mao et al.) | R50 / CLIP-B | 35.33 | 33.95 | 35.74 | 37.19 | 35.27 | 37.77 | 66.3 |
| LOGICHOI (Li et al.) | R50 / CLIP-B | 35.47 | 32.03 | 36.22 | 38.21 | 35.29 | 39.03 | 65.6 |
| DP-ADN (Gao et al.) | R50 / CLIP-B | 35.91 | 35.82 | 35.94 | 38.99 | 39.61 | 38.80 | 64.8 |
| HORP (Geng et al.) | R50 / CLIP-L | 38.61 | 36.14 | 39.34 | 40.98 | 38.25 | 41.79 | 68.3 |
| InterProDa (Jia et al.) | R50 / CLIP-L | 42.67 | 45.21 | 41.92 | – | – | – | – |
| DebiaHOI (Yang et al.) | R50 / CLIP-L | 42.93 | 42.41 | 43.11 | 44.97 | 44.20 | 45.23 | 72.1 |
| PViC (Zhang et al.) | Swin-L / – | 44.32 | 44.61 | 44.24 | 47.81 | 48.38 | 47.64 | 68.0 |
| MP-HOI (Yang et al.) | Swin-L / CLIP-L+SD | 44.53 | 44.48 | 44.55 | – | – | – | – |
| HORP (Geng et al.) | Swin-L / CLIP-L | 47.53 | 46.81 | 47.74 | 51.24 | 50.78 | 51.38 | 71.1 |
| *Zero-shot oriented methods* | | | | | | | | |
| CMMP (Lei et al.) | R50 / CLIP-B | 33.24 | 32.26 | 33.53 | 36.32 | 34.87 | 36.75 | 61.2 |
| ADA-CM (Lei et al.) | R50 / CLIP-B | 33.80 | 31.72 | 34.42 | 37.06 | 35.43 | 37.55 | 61.5 |
| EZ-HOI (Lei et al.) | R50 / CLIP-B | 33.15 | 29.11 | 34.36 | 36.38 | 31.93 | 37.71 | 63.5 |
| HOLa (Lei et al.) | R50 / CLIP-B | 35.41 | 34.35 | 35.73 | 38.59 | 36.43 | 39.10 | – |
| LAIN (Kim et al.) | R50 / CLIP-B | 36.02 | 35.70 | 36.11 | 39.17 | 38.16 | 39.47 | 65.1 |
| **LINK** | R50 / CLIP-B | **37.43** | **37.18** | **37.50** | **40.46** | **40.30** | **40.51** | **66.5** |
| CMMP (Lei et al.) | R50 / CLIP-L | 38.14 | 37.75 | 38.25 | 40.93 | 40.68 | 41.16 | 64.0 |
| ADA-CM (Lei et al.) | R50 / CLIP-L | 38.40 | 37.52 | 38.66 | 41.25 | 40.41 | 41.50 | 64.0 |
| EZ-HOI (Lei et al.) | R50 / CLIP-L | 38.61 | 37.70 | 38.89 | 41.65 | 40.75 | 41.91 | 65.4 |
| HOLa (Lei et al.) | R50 / CLIP-L | 39.05 | 38.66 | 39.17 | 42.13 | 41.18 | 42.42 | 66.0 |
| **LINK** | R50 / CLIP-L | **42.92** | **45.03** | **42.20** | **45.79** | **47.00** | **45.67** | **68.1** |
| UniHOI (Cao et al.) | R50 / BLIP-2-OPT-2.7B | 40.06 | 39.91 | 40.11 | 42.20 | 42.60 | 42.08 | 68.3 |
| BC-HOI (Hu et al.) | R50 / BLIP-2-OPT-2.7B | 43.01 | 45.76 | 42.18 | 45.35 | **47.94** | 44.57 | **70.6** |
| **LINK** | R50 / BLIP-2-OPT-2.7B | **43.72** | **45.82** | **43.10** | **46.11** | 47.71 | **45.62** | 68.5 |
| CMMP (Lei et al.) | Swin-L / CLIP-L | 44.26 | 45.48 | 43.89 | 47.15 | 48.36 | 46.79 | 65.5 |
| ADA-CM (Lei et al.) | Swin-L / CLIP-L | 44.99 | 45.98 | 44.69 | 47.77 | 49.08 | 47.38 | 65.7 |
| EZ-HOI (Lei et al.) | Swin-L / CLIP-L | 45.22 | 46.15 | 44.94 | 47.63 | 48.03 | 47.51 | 66.1 |
| HOLa (Lei et al.) | Swin-L / CLIP-L | 36.17 | 34.39 | 36.70 | 38.48 | 36.32 | 39.13 | – |
| **LINK** | Swin-L / CLIP-L | **49.06** | **53.63** | **47.60** | **51.34** | **56.29** | **49.86** | **69.2** |

Table 5: **Comparison on SWiG-HOI**, demonstrating the open-vocabulary capability.

| Method | N-rare | Rare | Novel | Full |
|---|---|---|---|---|
| Wang (Wang et al., 2021a) | 10.93 | 6.63 | 2.64 | 7.98 |
| THID (Wang et al., 2022) | 17.67 | 12.82 | 10.04 | 13.26 |
| AMP-HOI (Xue et al., 2024a) | 19.77 | 14.00 | 9.74 | 14.29 |
| MP-HOI-S (Yang et al., 2024) | 20.28 | 14.78 | - | 12.61 |
| GEN-VLKT (Liao et al., 2022) | 20.91 | 10.41 | - | 10.87 |
| CMD-SE (Lei et al., 2024b) | 21.46 | 14.64 | 10.70 | 15.26 |
| SGC-Net (Lin et al., 2025) | 23.67 | 16.55 | **12.46** | 17.20 |
| **LINK (ours)** | **24.37** | **17.88** | 12.15 | **17.97** |

highlight the strong generalization ability of our method to unseen HOI categories and validate its robustness in open-world scenarios.

**Few-shot Results.** We further evaluate few-shot performance on HICO-DET and V-COCO against state-of-the-art methods, as illustrated in Figure 3. Our method consistently achieving the best results across both benchmarks from 1-shot to 32-shot settings. Interestingly, we observe a trade-off in prior works: fully-supervised method PViC performs well on V-COCO but lags on HICO-DET, whereas zero-shot-oriented method ADA-CM† shows the reverse trend. We attribute this inconsistency to dataset scale differences that V-COCO contains only 24 HOI categories, while HICO-DET comprises 600. In contrast, our model maintains strong performance across both datasets.

**Comparison under Fully-Supervised Settings.** Main results on **HICO-DET and V-COCO** Benchmarks are presented in Table 4. On HICO-DET, our method achieves state-of-the-art performance across different model scales, including R50+ViT-B, R50+ViT-L, and Swin-L+ViT-L. Our small-scale model achieves 37.43 and 37.18 mAP on the Full and Rare subsets, respectively, outperforming the previous best query-free method (33.80 / 31.72 mAP) with relative improvements of 10.7% and 17.2%. Notably, while retaining the flexibility benefits of query-free designs, our method

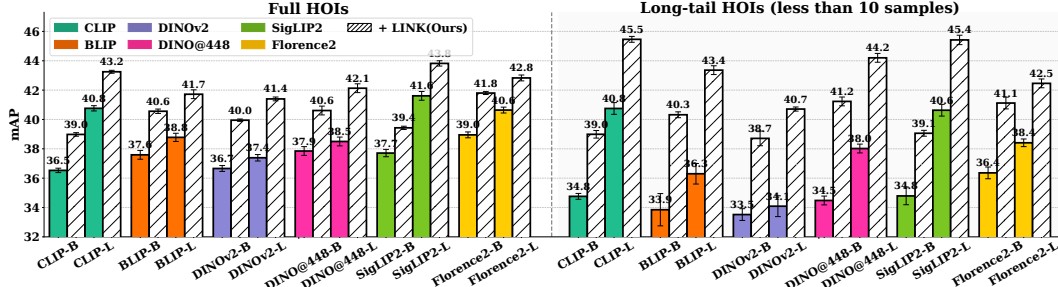

Figure 4: **Experiments on Diverse Foundation Models.** Our method (**+LINK**, striped bars) consistently improves the performance of all baselines with various foundation models: including **contrastive vision-language learning**-CLIP and BLIP; **self-supervised learning**-DINOv2 and DINO@448; and **multitask multimodal pretraining**-SigLIP2 and Florence2. Notably, LINK provides the largest gains on long-tail HOIs ($\leq 10$ samples), highlighting its generalization capability.

Table 6: Ablation on HICO-DET under fully-supervised setting.

| # | Encoder | Decoder | Full | Rare | N-Rare |
|---|---|---|---|---|---|
| A1 | Self-Attn | Cross-Attn | 36.10 | 33.67 | 36.97 |
| A2 | Self-Attn | VLM-Link | 39.23 | 39.76 | 39.02 |
| A3 | Geometrical | Cross-Attn | 38.30 | 35.46 | 39.31 |
| A4 | Geometrical | VLM-Link | 41.20 | 41.43 | 41.13 |
| A5 | + Logit-level KD | | 41.89 | 43.82 | 41.27 |
| A6 | + Query-level KD | | 42.34 | 43.62 | 41.84 |
| A7 | + Map-level KD | | 42.92 | 45.03 | 42.20 |
| A8 | + multi-teacher (CLIP + SigLIP) | | 43.54 | 45.58 | 42.93 |

Table 7: Ablation under zero-shot RF-UC setting.

| # | Encoder | Decoder | HM | Unseen | Seen | Full |
|---|---|---|---|---|---|---|
| A1 | Self-Attn | Cross-Attn | 31.74 | 30.23 | 33.36 | 32.54 |
| A2 | Self-Attn | VLM-Link | 35.47 | 34.78 | 36.17 | 35.84 |
| A3 | Geometrical | Cross-Attn | 34.55 | 33.54 | 35.60 | 35.10 |
| A4 | Geometrical | VLM-Link | 36.89 | 36.21 | 37.60 | 37.32 |
| A5 | + Logit-level KD | | 37.77 | 37.24 | 38.31 | 38.10 |
| A6 | + Query-level KD | | 39.10 | 38.30 | 39.92 | 39.48 |
| A7 | + Map-level KD | | 39.40 | 38.51 | 40.33 | 39.97 |

surpasses even all specific-query based methods on standard benchmarks. When scaling up to Swin-L+ViT-L, our model achieves 49.06 mAP, a +1.53 mAP gain over the prior best (HORP 47.53), highlighting excellent scalability. On V-COCO, our method also achieves competitive performance and outperforms prior two-stage methods, including ADA-CM (Lei et al., 2023) and PViC (Zhang et al., 2023) by a large margin.

**Comparison on Open-Vocabulary benchmark SWiG-HOI.** As shown in Table 5, our method achieves a new SOTA with 17.97 mAP on the full set, outperforming the previous best by 2.71 mAP (a relative 17.8%). We also obtain the best results on both Non-Rare (24.37 mAP, +13.6%) and Rare (17.88 mAP, +22.1%) subsets. Notably, our method achieves 12.15 mAP on the novel HOIs. demonstrating strong generalization capability.

In summary, the above comparative experiments demonstrate that our method excels in both specialization (three standard benchmarks) and generalization (zero-shot and few-shot settings), without sacrificing one for the other.

**Ablation Studies.** We conduct ablation studies on HICO-DET under both fully-supervised and zero-shot RF-UC settings, as shown in Table 6 and Table 7. Notably, Our baseline (A1) adopts a standard self-attention encoder over ROI features and a cross-attention decoder that attends to VLM representations, forming a plain baseline. First, we evaluate the impact of our architectural design. Introducing the HO Geometrical Encoder (A3) or VLM Linking decoder (A2) each improves performance over the baseline (A1), while combining both (A4) yields the best results, confirming their complementary benefits. Second, we investigate our instance-level knowledge learning strategy. Progressive integration of logit-level (A5), query-level (A6), and map-level (A7) distillation further boosts performance. Finally, employing multiple teachers (A8) achieves the highest gain in the fully-supervised setting, while in the zero-shot setting, instance-level learning still provides improvement, raising the harmonic mean by +2.51. These results demonstrate the effectiveness of both our architectural components and the proposed knowledge learning strategy.

**Various Foundation Models.** Beyond CLIP, we further evaluate our method across a diverse set of foundation models with different pre-training paradigms, including contrastive learning (BLIP), vision-only self-supervised learning (DINOv2, DINO@448), and multitask multimodal pre-training

(SigLIP2, Florence2). As illustrated in Figure 4, our method successfully adapts to all these models within a unified architecture, establishing strong baselines (solid bars in different colors). Furthermore, by incorporating our proposed learning strategy, performance is consistently and significantly improved (striped bars). Notably, +LINK enhances the detection of rare interactions, an essential capability for real-world HOI applications where long-tail categories are common and critical.

Table 8: Comparison of various OVD models on HICO-DET (Default and Known-Object settings).

| Open-vocabulary Object Detectors | Scale | HICO-DET (Default) | | | HICO-DET (KO) | | |
|---|---|---|---|---|---|---|---|
| | | Full | Rare | N-rare | Full | Rare | N-rare |
| YOLOv8-worldv2 | Small | 27.07 | 29.83 | 26.24 | 30.06 | 32.37 | 29.37 |
| YOLOv8-worldv2 | Medium | 30.35 | 33.40 | 29.45 | 33.39 | 36.00 | 32.62 |
| YOLOv8-worldv2 | Large | 32.64 | 36.51 | 31.49 | 35.98 | 40.06 | 34.76 |
| YOLOv8-worldv2 | XLarge | 33.73 | 37.30 | 32.66 | 36.98 | 40.65 | 35.88 |
| Grounding-DINO | Swin-tiny | 32.97 | 38.21 | 31.40 | 35.96 | 40.76 | 34.52 |
| Grounding-DINO | Swin-base | 39.69 | 45.99 | 37.81 | 42.42 | 47.94 | 40.77 |
| Qwen3-VL | 8B | 39.61 | 45.05 | 37.99 | 42.02 | 47.37 | 40.42 |

**Plug-and-Play with OVDs.** LINK operates as a detector-agnostic plug-and-play module over arbitrary open-vocabulary detectors (OVDs), requiring no retraining or adaptation of the underlying detector. Given bounding boxes from any OVD, LINK performs HOI reasoning independently of detector-specific representations. As shown in Table 8, integrating LINK with off-the-shelf OVDs, including YOLOv8-Worldv2, Grounding-DINO, and Qwen3-VL, yields strong and consistent performance across model scales.

## 5 CONCLUSION

In this paper, we presented a unified HOI detection framework that combines a HO Geometrical Encoder with a VLM Linking decoder, enabling seamless integration with diverse foundation models and object detectors while ensuring consistent performance across different scenarios. To address the limitations of sparse annotations, we introduced an instance-level knowledge learning strategy under a teacher-student paradigm, which provides dense and adaptive supervision across all humanobject pairs. Extensive experiments on HICO-DET, V-COCO, and SWiG-HOI demonstrate that our method achieves state-of-the-art performance and generalizes well across settings. Importantly, +LINK substantially improves the recognition of rare interactions, highlighting its value for real-world HOI applications where long-tail categories are both common and critical.

## ACKNOWLEDGEMENT

This work is supported by the research fund under Grant No. 20242910035 from the Tsinghua University-Jiangsu CRRC Digital Technology Co., Ltd. Joint Research Center for Data Driven Intelligence of Industry.

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

# A APPENDIX

## A.1 OVERVIEW OF THE PROPOSED LINK

To develop a VLM-based HOI detector that performs robustly across both specific and generalizable scenarios, our work is guided by two central questions: *How can we design a unified and universal HOI detection architecture that maintains consistent performance across different settings? How can we generate dense and informative supervision signals to better guide HOI learning, especially under sparse annotations?* To this end, we propose the **Learning Instance-level Knowledge** framework (LINK), a generalizable and modular architecture for HOI detection built upon vision-language models (VLMs). As illustrated in Fig. 2 of the main paper, LINK consists of two key components: a **HO Geometrical Encoder** that models pairwise spatial relationships between humans and objects to enhance contextual reasoning, and a **VLM-Linking Decoder** that fuses native and latent representations for robust HOI prediction across diverse tasks. On top of this architecture, we introduce an **Instance-level Knowledge Learning** strategy that adopts a fully mirrored teacher-student paradigm. This design enables multi-level supervisionspanning features, queries, and logitsover *all* candidate human-object pairs. Unlike traditional methods that only supervise ground-truth matches, our strategy generates adaptive and learnable signals to distinguish both positive and negative instances at fine granularity. Together, this unified design ensures plug-and-play compatibility with different object detectors and foundation models, while our dense supervision paradigm enhances generalization to rare, zero-shot, and open-vocabulary settings. The proposed LINK framework thus provides a scalable, transferable, and high-performance solution for modern HOI detection.

## A.2 DATASETS FOR HOI DETECTION

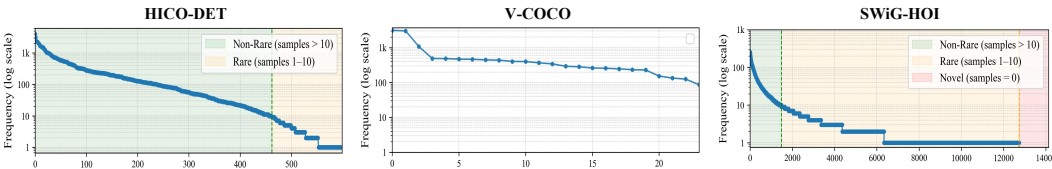

Figure 5: Distribution of widely-used HOI datasets: HICO-DET, V-COCO and SWiG-HOI.

Our experiments are conducted on three widely used HOI detection benchmarks: HICO-DET, V-COCO, and SWiG-HOI. Below, we provide a detailed overview of each dataset in terms of composition and scale.

**HICO-DET** is an HOI detection benchmark extended from the HICO classification dataset. It includes 600 HOI categories, formed from 80 object categories and 117 verb categories. The dataset contains a total of 47,100 images (37.6k for training and 9.5k for testing), with approximately 151.2k annotated interaction instances (117.8k for training and 33.4k for testing). It supports diverse interaction modeling and exhibits a clear long-tail distribution139 of the 600 HOI categories have fewer than 10 training samples. Evaluation is typically conducted under three settings: Full (all 600 categories), Rare (long-tail HOIs), and Non-Rare (the remaining categories).

**V-COCO** is a smaller-scale HOI dataset built on MS-COCO Lin et al. (2014), focusing on recognizing common actions. It defines 24 HOI categories involving 80 object categories and 24 verbs. The dataset consists of 10.3k images (5.4k for training and 4.9k for testing) and approximately 26.2k annotated interactions (13.8k for training and 12.4k for testing).

**SWiG-HOI** is a large-vocabulary HOI benchmark derived from the SWiG and DOH datasets. It features a highly diverse interaction space and open-vocabulary combinations, making it suitable for evaluating generalization and long-tail performance. It defines 14,130 HOI categories from 1,000 object categories and 407 verb categories. The dataset includes 54.6k training images and 13.6k test images, with a total of 99.8k annotated interaction instances (80.2k for training and 19.6k for testing).

Overall, HICO-DET serves as a widely adopted benchmark; V-COCO offers compact annotations in standardized COCO scenes; and SWiG-HOI is designed for large-scale open-world evaluation. Our method, with its unified architecture, is capable of adapting to diverse scenariosincluding open-

| Detector | Ours | | | PViC (Zhang et al., 2023) | | |
|---|---|---|---|---|---|---|
| | Full | Rare | N-rare | Full | Rare | N-rare |
| DETR-R50* | 43.54 | 45.58 | 42.93 | 34.32 | 31.62 | 35.13 |
| Deformable-DETR-R50 | 41.66 | 45.65 | 40.46 | 18.72 | 13.49 | 20.28 |
| H-Deformable-DETR-SwinL | 49.06 | 53.63 | 47.60 | 15.83 | 10.94 | 17.29 |
| YOLO$_{v12}$-nano | 27.12 | 25.77 | 27.52 | 14.42 | 10.72 | 15.51 |
| YOLO$_{v11}$-X | 30.62 | 28.46 | 31.13 | 13.63 | 9.90 | 14.75 |

Table 9: **Cross-detector evaluation on HICO-DET.** Both methods are trained with DETR-R50 (*). Our method generalizes well to new detectors without fine-tuning, while PViC degrades significantly, showing its reliance on detector-specific features.

vocabulary, zero-shot, few-shot, and low-label settings. Its effectiveness has been thoroughly evaluated across all three datasets.

## A.3 ADAPTABILITY TO OBJECT DETECTORS

Thanks to our architecture's fully decoupled design, LINK is inherently adaptable to a wide range of object detectors. Specifically, our model does not rely on any detector-specific features or intermediate representations. Instead, it solely requires bounding boxes as input and performs spatial encoding based on the geometry of human-object pairs.

This design enables **plug-and-play compatibility** with arbitrary object detectors, such as Faster R-CNN, YOLO, DETR, Deformable DETR or even visual grounding MLLMs, without requiring any fine-tuning or feature alignment. As a result, LINK provides a flexible and efficient solution that can be readily deployed across different detection backbones and application domains, while maintaining consistent performance and minimizing adaptation overhead.

Table 9 presents a cross-detector evaluation on the HICO-DET dataset to assess the adaptability of our architecture. The results compare our method (top) and PViC Zhang et al. (2023) (bottom) under five different object detectors, including DETR-R50*, Deformable DETR, H-Deformable DETR with Swin-L backbone, and two YOLO variants.

The rows marked with * indicate that both methods are trained using DETR-R50 as the object detector. When directly replacing the detector at inference timewithout any additional fine-tuningour method maintains robust performance across all detectors. This highlights the plug-and-play nature of our architecture, which is fully decoupled from detector-specific features and relies solely on bounding box inputs for interaction reasoning. In contrast, PViC and similar prior methods exhibit significant performance degradation when the detector is changed, with mAP dropping to nearly unusable levels. This illustrates their strong coupling with detector-specific features and the need for costly re-training whenever the detection backbone is modified.

These results demonstrate that our method offers superior flexibility and generalization, making it more practical for real-world deployment.

## A.4 MORE IMPLEMENTATION DETAILS

**Training Setup.** We build our implementation upon the official open-source codebase of PViC Zhang et al. (2023), extending it to support our proposed LINK framework. All experiments are conducted using PyTorch with 8 NVIDIA RTX 4090 GPUs (24GB each). Our method is hardware-friendly: we train the model for 15 epochs with a total batch size of 16, using 4 GPUs under Distributed Data Parallel (DDP) training. Following PViC, we adopt the same preprocessing and loading pipeline for the HICO-DET and V-COCO datasets. Additionally, we generalize the data interface and configuration system to support arbitrary datasets in the same manner, including the large-scale SWiG-HOI dataset introduced earlier.

**Hyper-parameters.** We adopt the Focal Loss for classification with two key parameters: `alpha = 0.5`, `gamma = 0.1`, controlling the loss weighting and focusing factor. We filter predicted instances using a confidence threshold of `box-score-thresh = 0.05`, and retain

a dynamic number of predictions per image within the range of `min-instances = 3` to `max-instances = 15`. In the zero-shot setting, we apply a top-k filtering strategy for candidate verbs and objects, with `zs-topk = 10` and a scaling factor `zs-topk-factor = 1.8` to dynamically adjust the number of retained interaction candidates based on instance confidence.

**Loss Details.** As described in the main paper, our training objective combines a standard focal loss for HOI classification with a set of auxiliary losses designed to support instance-level knowledge learning under a teacher-student paradigm. The overall objective is given by:

$$\theta^* = \arg\min_\theta \mathbb{E}_{\mathcal{I} \sim \mathcal{X}} \left[ \mathcal{L}_\mathcal{M}(\Phi_\theta(\mathcal{I}, \mathcal{B}), \mathcal{GT}) + \sum_{g \in \mathcal{G}} \mathcal{KD}_g(\Phi_\theta(\mathcal{I}, \mathcal{B}), \Phi_t(\mathcal{I}, \mathcal{B})) \right] \quad (8)$$

Here, $\mathcal{L}_\mathcal{M}$ denotes the focal loss applied to the main classification outputs, and $\mathcal{KD}_g$ represents a set of knowledge distillation objectives at different levels $g \in \mathcal{G}$, including logits, queries, and feature maps. Specifically, we assign the following weights to the instance-level auxiliary losses:

- `loss_logits`: 1.0   *(Ground Truth classification logits)*
- `loss_query_native`: 1.0   *(decoder native branch)*
- `loss_query_latent`: 1.0   *(decoder latent branch)*
- `loss_feat_map`: 0.5   *(intermediate feature maps)*
- `loss_query_encoder`: 0.5   *(encoder HO queries)*

In practice, we find that varying these weights does not significantly affect the model's final performance, indicating that our learning strategy is robust to hyperparameter settings. The chosen values are primarily aimed at balancing the relative magnitudes of each loss term, ensuring that no single component dominates the overall training objective.

These components collectively provide multi-level, instance-aware supervision across both encoder and decoder, helping the student model to more effectively distinguish fine-grained human-object interactions.

## A.5 COMPREHENSIVE COMPARISON WITH MULTIMODAL LARGE LANGUAGE MODELS

To enable a fair and reproducible comparison, we define a standardized evaluation protocol and output format for applying Multimodal Large Language Models (MLLMs) to the HOI detection task. Since MLLMs typically produce textual outputs, we explicitly define both the input-output structure and formatting rules for extracting HOI predictions.

**Output Format Definition.** The MLLM is expected to produce a structured output in the following format:

```
HOI_result = {
  "boxes": [...],        # list of [x1, y1, x2, y2]
  "classes": [...],      # class IDs aligned with boxes
  "interactions": [      # list of HOI triplets
    {"human_idx": int, "object_idx": int, "verb": int},
    ...
  ]
}
```

**Formatting Rules:**

- Indices `human_idx` and `object_idx` refer to entries in the `boxes` list.
- Class ID `0` corresponds to `human`; others indicate object categories.
- For multiple verbs, use multiple entries with shared indices and distinct `verb` IDs.
- The output must be JSON-compatible.

We observe that directly prompting MLLMs to perform object detection and HOI predictioneither through supervised fine-tuning or in a zero-shot settingremains highly challenging due to their limited spatial grounding capabilities. To address this, we provide MLLMs with the object detectors predictions (i.e., bounding boxes and class labels) as part of the input prompt. This strategy enables a

| Method | mF1 | | | mPrec | | | mRec | | |
|---|---|---|---|---|---|---|---|---|---|
| | Full | Rare | N.Rare | Full | Rare | N.Rare | Full | Rare | N.Rare |
| **HOI Detection Based on R50-DETR OB Detection Prompting** | | | | | | | | | |
| Qwen-vl-max | 0.1719 | 0.1523 | 0.1777 | 0.2030 | 0.1585 | 0.2164 | 0.1833 | 0.1546 | 0.1919 |
| Claude-3.5-sonnet | 0.1895 | 0.2429 | 0.1736 | 0.2071 | 0.2553 | 0.1927 | 0.2099 | 0.2554 | 0.1963 |
| GPT-o4-mini | 0.2082 | 0.2058 | 0.1958 | 0.2271 | 0.2120 | 0.2173 | 0.2419 | 0.2186 | 0.2337 |
| GPT-o3 | 0.2758 | 0.3310 | 0.2592 | 0.2792 | 0.3507 | 0.2579 | 0.3328 | 0.3430 | 0.3297 |
| Gemini-2.5-flash | 0.2680 | 0.3347 | 0.2481 | 0.2754 | 0.3412 | 0.2557 | 0.3305 | 0.3659 | 0.3199 |
| Gemini-2.5-pro | 0.3161 | 0.4010 | 0.2908 | 0.3084 | 0.4145 | 0.2767 | 0.3937 | 0.4287 | 0.3833 |
| CMMP (0.48B) | 0.3324 | 0.3024 | 0.3414 | 0.3324 | 0.3024 | 0.3414 | 0.4352 | 0.3448 | 0.4622 |
| ADA-CM (0.44B) | 0.3414 | 0.3133 | 0.3498 | 0.3225 | 0.3181 | 0.3238 | 0.4469 | 0.3611 | 0.4715 |
| **Ours (0.52B)** | 0.3671 | 0.3542 | 0.3710 | 0.3351 | 0.3538 | 0.3296 | 0.4982 | 0.4149 | 0.5231 |
| **HOI Detection Based on Ground Truth OB Detection Prompting** | | | | | | | | | |
| Qwen2.5-vl-72b | 0.2338 | 0.2486 | 0.2294 | 0.2801 | 0.2681 | 0.2836 | 0.2317 | 0.2440 | 0.2281 |
| Qwen-vl-max | 0.2930 | 0.2605 | 0.3027 | 0.3320 | 0.2754 | 0.3489 | 0.2935 | 0.2606 | 0.3033 |
| Claude-3.5-haiku | 0.1000 | 0.0871 | 0.1038 | 0.1156 | 0.1001 | 0.1202 | 0.1302 | 0.1025 | 0.1384 |
| Claude-3.5-sonnet | 0.2652 | 0.3145 | 0.2505 | 0.2815 | 0.3333 | 0.2660 | 0.3019 | 0.3405 | 0.2904 |
| Claude-4-sonnet | 0.2485 | 0.2303 | 0.2539 | 0.2619 | 0.2440 | 0.2673 | 0.2844 | 0.2473 | 0.2954 |
| Claude-4-opus | 0.3044 | 0.2931 | 0.3077 | 0.3232 | 0.3103 | 0.3271 | 0.3378 | 0.3146 | 0.3447 |
| GPT-4o-mini | 0.1306 | 0.1881 | 0.1134 | 0.1450 | 0.1976 | 0.1292 | 0.1455 | 0.2007 | 0.1290 |
| GPT-4o | 0.2832 | 0.3334 | 0.2590 | 0.3083 | 0.3490 | 0.2861 | 0.3162 | 0.3425 | 0.2981 |
| GPT-o4-mini | 0.3631 | 0.4111 | 0.3463 | 0.3943 | 0.4236 | 0.3830 | 0.3791 | 0.4360 | 0.3596 |
| GPT-o3 | 0.4676 | 0.5152 | 0.4533 | 0.4657 | 0.5287 | 0.4469 | 0.5328 | 0.5607 | 0.5244 |
| Gemini-2.0-flash | 0.2708 | 0.3020 | 0.2615 | 0.3026 | 0.3126 | 0.2997 | 0.2866 | 0.3200 | 0.2767 |
| Gemini-1.5-pro | 0.2933 | 0.3854 | 0.2595 | 0.3291 | 0.3853 | 0.3053 | 0.3084 | 0.4060 | 0.2726 |
| Gemini-2.5-flash | 0.4224 | 0.4697 | 0.4082 | 0.4548 | 0.4815 | 0.4469 | 0.4475 | 0.4963 | 0.4330 |
| Gemini-2.5-pro | 0.4623 | 0.4977 | 0.4517 | 0.5044 | 0.5377 | 0.4944 | 0.4797 | 0.5150 | 0.4692 |
| CMMP (0.48B) | 0.4476 | 0.4648 | 0.4425 | 0.4077 | 0.4515 | 0.3947 | 0.6032 | 0.5742 | 0.6119 |
| ADA-CM (0.44B) | 0.4581 | 0.4257 | 0.4678 | 0.4197 | 0.4017 | 0.4251 | 0.6139 | 0.5303 | 0.6389 |
| **Ours (0.52B)** | 0.5009 | 0.5342 | 0.4910 | 0.4667 | 0.5185 | 0.4512 | 0.6443 | 0.6243 | 0.6502 |

Table 10: Comparison with MLLMs. Higher is better. Top-3 values in each column are color-coded.

fair comparison between MLLM-based models and traditional two-stage HOI detectors, by aligning their input structure and evaluation procedure.

By clearly defining the task, unifying the input format, and standardizing the output representation, we make it possible to evaluate MLLMs on HOI detection in a structured and consistent manner. This also allows structured predictions to be extracted from free-form text generated by different MLLMs, making quantitative comparisons across models both feasible and fair.

**Evaluation Metrics** Since MLLMs generate textual outputs rather than full confidence scores over all possible categories, traditional metrics such as mean Average Precision (mAP) are not well-suited for evaluation in this setting. Therefore, we adopt three alternative metrics: **mean F1 score (mF1)**, **mean precision**, and **mean recall** to better reflect the accuracy of MLLM-based HOI predictions. We follow the standard matching logic used in prior HOI works: a predicted human-object interaction is considered a true positive if and only if:

- Both the predicted human and object bounding boxes have an Intersection-over-Union (IoU) greater than 0.5 with the corresponding ground truth boxes;
- The predicted object category and interaction (verb) label match the ground truth.

This ensures a fair and consistent evaluation for both MLLM-based and conventional HOI detection methods.

**Results.** We evaluate a series of models under two settings: (1) R50-DETR Object Detection Prompting, where detected boxes and class labels from R50-DETR are provided as prompts; and (2) Ground Truth (GT) Object Detection Prompting, where oracle boxes and class labels are used. The compared models include MLLMs (e.g., Qwen, GPT, Gemini, Claude series) and HOI-specific methods (e.g., CMMP, ADA-CM, Ours). In the **R50-DETR prompting setting**, ours achieves the highest mF1 score (36.71%), outperforming both the sota HOI method baseline (CMMP, 33.24%) and all MLLMs (best being Gemini-2.5-pro, 31.61%). Similar trends are observed for mPrecision and mRecall, where our method achieves 33.51% and 49.82%, respectively. This demonstrates that our method is more robust under detection noise and better captures interaction semantics even

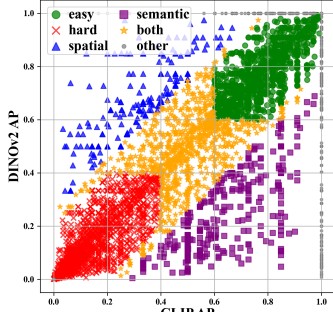

| Full | Spat. | Sema. | Both | Hard |
|---|---|---|---|---|
| DINOv2-large – *Baseline v.s +LINK* | | | | |
| 37.39 | 65.7 | 24.4 | 48.8 | 16.4 |
| 41.40↑ | 58.1↓ | 46.5↑ | 51.7↑ | 19.4↑ |
| CLIP-large – *Baseline v.s +LINK* | | | | |
| 41.2 | 31.4 | 63.0 | 49.3 | 17.0 |
| 43.3↑ | 42.9↑ | 61.4↓ | 51.7↑ | 19.1↑ |
| Florence2-Base – *Baseline v.s +LINK* | | | | |
| 38.9 | 50.9 | 48.1 | 51.8 | 19.9 |
| 41.8↑ | 51.1↑ | 51.3↑ | 52.5↑ | 20.3↑ |

Figure 6: Spatial *vs.* Semantic emphasis in foundation models for HOI detection and the impact of our method.

when the input object detections are imperfect. In the **GT prompting setting**, which isolates the HOI prediction capability by removing detection errors, our method again achieves the highest performance: mF1 of 50.09%, mPrecision of 46.67%, and mRecall of 64.43%. These results exceed not only MLLMs (e.g., GPT-4o-mini at 36.31%, Gemini-2.5-pro at 46.23%) but also prior HOI methods like CMMP (44.76%) and ADA-CM (45.81%).

In particular, our method shows superior performance on Rare interactions (mF1-Rare: 53.42%) and Non-Rare interactions (mF1-N.Rare: 49.10%), indicating both generalization ability and capacity for modeling long-tail HOI categories. Compared to other HOI methods, our approach yields consistent improvements across all splits. Overall, these results validate the effectiveness of our unified HOI detection framework, outperforming both MLLMs and previous dedicated HOI approaches under both detection-prompted and oracle-prompted settings.

## A.6   SPATIAL AND SEMANTIC EMPHASES

**Spatial *vs.* Semantic Emphasis.**   To further assess the impact of our proposed LINK module, we investigate how foundation models vary in their spatial and semantic emphasis within a unified HOI detection architecture. Specifically, we compare two representative baselines: CLIP-large, which is semantic-oriented due to its image-text contrastive pretraining, and DINOv2-large, which is spatial-oriented via visual-only self-supervised learning.

On the 9.6K test images (covering 33K HOI instances), we categorize samples into four subsets based on the normalized performance difference between CLIP and DINOv2: (1) *Semantic-Oriented* (CLIP − DINOv2 > 0.25), (2) *Spatial-Oriented* (DINOv2 − CLIP > 0.25), (3) *Both* (|CLIP − DINOv2| ≤ 0.25), and (4) *Hard* cases (CLIP < 0.4 and DINOv2 < 0.4).

As shown in Figure 6, DINOv2 achieves strong performance on spatial dominated cases (Spat.: 65.7) but performs poorly on semantic-heavy samples (Sema.: 24.4), revealing a strong spatial bias and limited capacity for high-level semantic reasoning. After integrating our LINK module, this disparity is greatly mitigated (Spat.: 58.1 vs. Sema.: 46.5), resulting in enhanced generalization across categories and a clear improvement in overall mAP (from 37.4 to 41.4).

In contrast, CLIP initially demonstrates a semantic preference (Sema.: 63.0 vs. Spat.: 31.4). With the addition of LINK, spatial sensitivity is substantially improved (Spat.: 42.9), while semantic strength remains largely preserved (Sema.: 61.4), leading to balanced capability and improved performance on hard samples (17.0 → 19.1).

Even for Florence2-Base, which is pretrained using multimodal multitask learning and exhibits more balanced behavior, LINK continues to deliver consistent improvements across all subsetsspatial, semantic, both, and harddemonstrating its universal adaptability and effectiveness in addressing spatial-semantic imbalance in diverse foundation models.

## A.7   VISUALIZATION RESULTS

To provide a more intuitive understanding of our models behavior, we present qualitative results of HOI detection on both real and synthetic images in Figure 7.

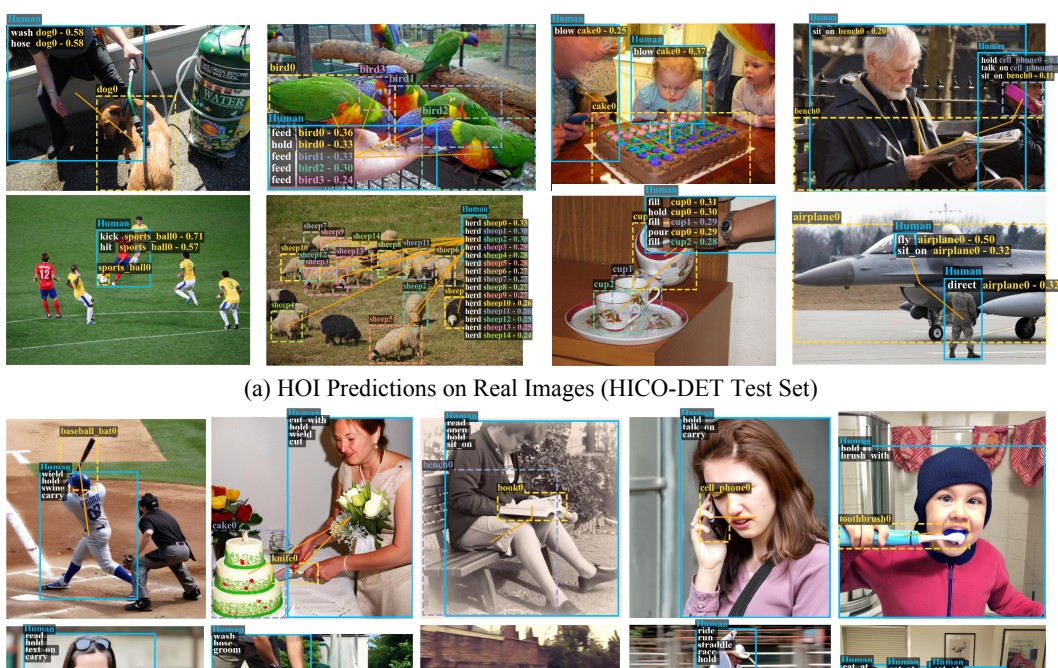

(a) HOI Predictions on Real Images (HICO-DET Test Set)

(b) HOI Predictions on Synthetic Images (Cross-domain Zero-shot Setting)

Figure 7: Visualization of HOI predictions on both real and synthetic images. (a) HOI detection results on real images from the HICO-DET test set demonstrate strong instance awareness and precise interaction reasoning. (b) Cross-domain zero-shot predictions on synthetic images highlight the models generalization ability and robustness to visual domain shifts.

As shown in Figure 7(a), on real-world test images from HICO-DET, our method demonstrates strong instance awareness and precise HOI reasoning. For example, in the second image of the first row, our model accurately identifies the action `feed` for all relevant bird instances (bird0bird4), while also distinguishing the `hold` action exclusively for bird0. Notably, it correctly predicts `no interaction` for the background birds, showcasing its ability to selectively recognize meaningful human-object pairs amidst complex scenes. In the fourth image of the same row, our model successfully disambiguates multiple overlapping interactions involving the same object. It predicts that both the man and the woman are `sit on` the `bench`, while additionally identifying the womans action of `hold` and `talk on cellphone` as a distinct HOI triplet. This demonstrates the models capacity to capture layered and concurrent human-object relations in everyday scenes.

In Figure 7(b), we further demonstrate the models zero-shot generalization on synthetic and out-of-domain images. Despite visual domain shifts, our model continues to detect plausible HOI triplets (e.g., `ride horse`, `brush with toothbrush`, `use cell phone`) with high fidelity, highlighting its cross-domain robustness and ability to generalize to unseen scenarios.

## A.8 CROSS-DOMAIN EVALUATION

In this section, we evaluate the cross-domain generalization ability of our method and compare it against previous methods. As illustrated in Table 11, we consider multiple domain shift settings, where the domain on the left side of the arrow indicates the training domain, and the right side denotes the testing domain. Specifically, **SWiG**, **H**, and **V** refer to the SWiG-HOI, HICO-DET, and V-COCO datasets, respectively. For example, *SWiG → HICO* indicates that the model is trained

| Method | SWiG→HICO-DET | SWiG→V-COCO | HICO-DET→V-COCO | Real→Synthetic |
|---|---|---|---|---|
| PViC | 9.73 / 13.66 | 38.2 | 45.1 | 33.70 / 31.32 |
| MP-HOI | — / — | 44.2 | — | — / — |
| ADA-CM | — / — | — | 46.6 | 29.87 / 28.17 |
| CMMP | — / — | — | 47.2 | 29.43 / 28.52 |
| **Ours** | **26.64 / 30.95** | **51.2** | **52.8** | **35.91 / 34.91** |

Table 11: Cross-domain HOI detection results. Each entry reports mAP scores (Full / Rare when available) on the target domain without fine-tuning. Our method consistently outperforms prior works across all cross-domain settings.

on SWiG-HOI and directly evaluated on HICO-DET without any fine-tuning. The setting *Real → Synthetic* evaluates the model trained on real-world images and tested on synthetic/generated ones, with partial visualization results presented in the previous section (Figure 7).

These results highlight the superior cross-domain These results underscore the strong cross-domain generalization capability of our approach. Across all evaluated settings, our method achieves substantial improvements over prior works. Notably, in the *SWiG→HICO* setting, our model significantly outperforms PViC (26.64 vs. 9.73 on Full, and 30.95 vs. 13.66 on Rare), demonstrating the high transferability and generality of the learned representations.

In the *HICO→V* setting, despite both datasets containing real-world images with differing verb and object distributions, our model achieves 52.8 mAP, surpassing the previous best of 47.2 by CMMP. This highlights the effectiveness of our HO Geometrical encoder and VLM Linking decoder in capturing robust, transferable features across varying interaction patterns. Furthermore, in the *Real→Synthetic* scenariowhere substantial shifts in visual appearance, object style, and contextual cues existour model maintains strong performance (35.91 / 34.91), outperforming all baselines. These results collectively demonstrate the robustness and adaptability of our framework under diverse and challenging domain shifts.

**How to Generate Synthetic Images with Annotations.** In the evaluations reported in Table 11, we use the original test sets of SWiG-HOI, HICO-DET, and V-COCO for the respective settings. However, for the *Real → Synthetic* evaluation, we need to generate synthetic images with corresponding HOI annotations to enable quantitative assessment.

To this end, we adopt InteractDiffusion Hoe et al. (2024), a condition-controlled image generation framework based on Stable Diffusion that extends existing pre-trained text-to-image (T2I) diffusion models to better incorporate human-object interaction conditioning. Specifically, we use the InteractDiffusion model built upon the Stable Diffusion XL version. We take the bounding boxes from the HICO-DET test set as layout guidance and use their corresponding ground-truth HOI triplets as text prompts. These two signals, layout and text, are jointly fed into model to generate realistic images that reflect the specified interactions. This approach allows us to synthesize a new test set of images with known annotations, enabling systematic evaluation of HOI models under domain shifts from real to synthetic data.

### A.9 COMPUTATIONAL COMPLEXITY ANALYSIS

Our training pipeline involves an additional pre-training stage for a teacher model, followed by student training supervised by both ground-truth annotations and teacher guidance. While this design raises potential concerns regarding computational resources and training time, we empirically show that these costs remain within a reasonable range.

As illustrated in Figure 8, our method achieves competitive performance within just one epoch and reaches full convergence in 15 epochs, significantly faster than prior methods in terms of convergence speed. Thanks to this rapid convergence, our method requires fewer total epochs. Although the overall training time is slightly longer due to the inclusion of teacher supervision, the increase is acceptable given the performance gains. Moreover, our method maintains computational efficiency during inference. For instance, LINK-B achieves 37.43 mAP with only 80.1 GFLOPs. Since the teacher-student paradigm is used solely during training, it does not incur any additional cost at inference time, ensuring that the model remains efficient.

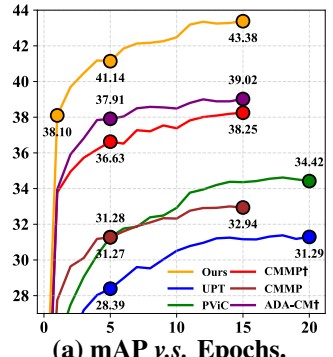

| Method | Perf | GFLOPs | Times |
|---|---|---|---|
| UPT | 31.66 | 60.9 | 3h47m |
| PViC | 34.69 | 62.1 | 4h02m |
| CMMP | 33.24 | 114.0 | 3h24m |
| CMMP† | 38.14 | 168.0 | 6h18m |
| LINK-B | 37.43 | 80.1 | 4h16m |
| LINK-B* | 39.43 | 79.9 | 4h56m |
| LINK-L | 43.25 | 252.5 | 6h48m |
| LINK-L* | 43.82 | 251.9 | 6h04m |
| *Teacher model require an additional one-time cost of 4h12m.* | | | |

**(a) mAP *v.s.* Epochs.**  **(b) Comparison with SOTAs.**

Figure 8: This illustrates the performance progression over epochs and the total time required for training. All experiments are conducted on four NVIDIA-RTX-4090 GPUs.

In summary, our method converges faster and delivers a favorable balance between training cost and final model efficiency.

### A.10 SCALABILITY

We visualize the performance of various methods across different computational scales in Figure 9, plotting mAP against GFLOPs on a logarithmic axis.

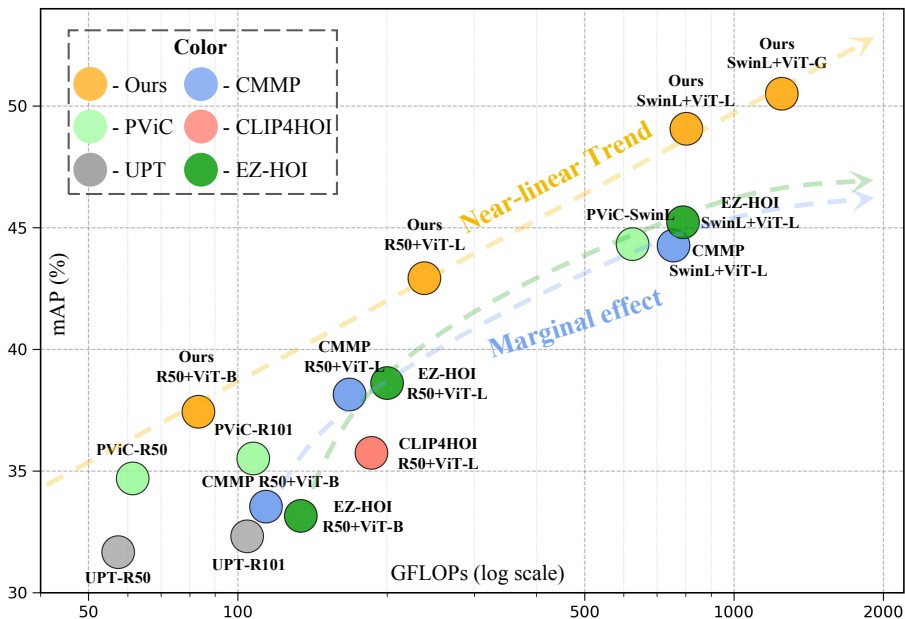

Figure 9: Our method exhibits nearly linear performance growth with respect to GFLOPs under the logarithmic scale, demonstrating excellent scalability. Moreover, across all model sizes, our method consistently achieves state-of-the-art performance compared with existing methods.

Our method demonstrates exceptional scalability, exhibiting an almost linear performance trend (yellow dashed arrow) as computational capacity increases under log-scale GFLOPs. When equipped with a Swin-Large backbone and further scaled via a ViT-Giant VLM, our LINK model reaches 50.5 mAP on HICO-DET dataset, markedly outperforming all existing methods.

In contrast, VLM-based methods such as CMMP and EZ-HOI display progressively diminishing performance gains as model size increases (blue/green dashed arrow), deviating substantially from

linear scaling and revealing clear marginal returns. Moreover, nonVLM-based pipelines such as UPT and PViC are unable to leverage large-scale pretrained models to effectively scale up.

These results collectively underscore the superior scaling characteristics of our method, enabling consistent and efficient utilization of expanding computational and representational capacity.

## A.11 PLUG-AND-PLAY WITH OPEN-VOCABULARY DETECTORS

| Open-vocabulary Detectors | HICO-DET (Default) | | | HICO-DET (Known-Object) | | | HICO → V-COCO |
|---|---|---|---|---|---|---|---|
| | Full | Rare | N-rare | Full | Rare | N-rare | $AP_{role}^{S2}$ |
| YOLO-world-s | 27.07 | 29.83 | 26.24 | 30.06 | 32.37 | 29.37 | 33.61 |
| YOLO-world-m | 30.35 | 33.40 | 29.45 | 33.39 | 36.00 | 32.62 | 37.31 |
| YOLO-world-l | 32.64 | 36.51 | 31.49 | 35.98 | 40.06 | 34.76 | 39.28 |
| YOLO-world-x | 33.73 | 37.30 | 32.66 | 36.98 | 40.65 | 35.88 | 40.35 |
| Grounding-DINO swin-tiny | 32.97 | 38.21 | 31.40 | 35.96 | 40.76 | 34.52 | 36.56 |
| Grounding-DINO swin-base | 39.69 | 45.99 | 37.81 | 42.42 | 47.94 | 40.77 | 46.55 |
| Qwen3-VL-8B | 39.61 | 45.05 | 37.99 | 42.02 | 47.37 | 40.42 | – |

Table 12: Comparison of various OVD models on HICO-DET (Default and Known-Object settings) and cross-dataset transfer from HICO-DET to V-COCO.

We conduct extensive evaluations across several state-of-the-art open-vocabulary detectors (OVDs), including YOLO-World (s/m/l/x), Grounding-DINO (Swin-Tiny/Base), and Qwen3-VL-8B. None of the detectors are fine-tuned on the target dataset, and our LINK model is also not adapted to any detector, making the entire pipeline fully training-free and plug-and-play.

For a fair comparison, we adopt the following configurations:

- **YOLO-World**: input resolution of 640 and score threshold of 0.25.
- **Grounding-DINO**: box threshold of 0.35 and text threshold of 0.25.
- **Qwen3-VL**: top_p=0.8, top_k=20, temperature=0.2, max_tokens=2048; outputs are normalized to $[0, 1000]$.

Despite being entirely training-free, our method achieves strong and consistent performance across all evaluated OVDs. Notably, combining LINK with **Grounding-DINO (Swin-Base)** even surpasses several approaches that rely on fine-tuned detectors.

We further evaluate the **HICO → V-COCO** transfer setting, where a LINK model trained on HICO-DET is directly paired with each OVD. Our method remains robust under this cross-dataset setting, demonstrating the scalability and strong generalization ability of our design.

We also present qualitative comparisons illustrating how different open-vocabulary detectors behave in real scenes and how LINK leverages their outputs for HOI reasoning. The first three columns visualize the raw detection results from Qwen3-VL-8B, YOLO-World-X, and GroundingDINO Swin-Base, respectively. These detectors often generate a large number of candidate boxes with heterogeneous confidence distributionsan inherent challenge when performing HOI reasoning in a fully training-free setting.

Despite these inconsistencies, the last column shows that LINK produces accurate and semantically coherent HOI predictions when directly paired with GroundingDINO Swin-Base, without any fine-tuning on either detector or HOI model. Notably, in the third row, the scene contains many irrelevant object proposals (e.g., multiple cars in the background and a baseball glove near the player), yet LINK successfully filters out distractors and identifies the correct human-object interaction. It outputs meaningful interaction labels such as hold, wield, and swing for the baseball bat, demonstrating the robustness and strong generalization capability of our method under noisy OVD detections.

## A.12 ROBUSTNESS UNDER INACCURATE DETECTED BOXES.

To evaluate architectural robustness, we introduce two forms of spatial perturbation: **box-shift** and **box-scale**. For **box-shift**, a bounding box with width $w$, height $h$, and center $(C_x, C_y)$ is perturbed by uniformly sampling a new center within $[C_x \pm \delta w]$ and $[C_y \pm \delta h]$. For **box-scale**, the width and height are randomly rescaled within the range $[(1 - \delta), (1 + \delta)]$. We evaluate $\delta \in [0, 1]$, and

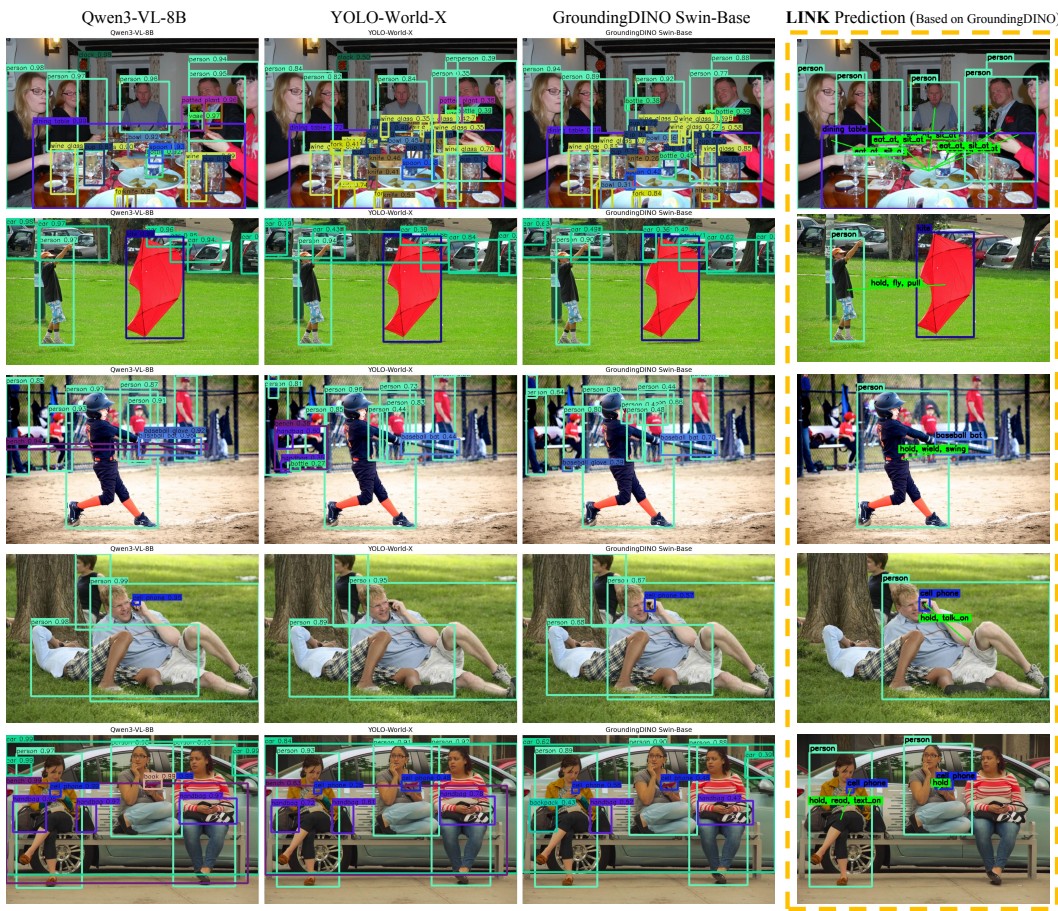

Figure 10: Qualitative results comparing OVD detections (first three columns) with HOI predictions produced by LINK using GroundingDINO detections (last column). Although OVDs output numerous proposals, LINK reliably identifies the correct human-object pairs and interactions.

| Perturbation Type | | Perturbation Strength | | | | | |
|---|---|---|---|---|---|---|---|
| | | 0.0 | 0.2 | 0.4 | 0.6 | 0.8 | 1.0 |
| **Box-Shift** | Full | 42.93 | 42.69 | 42.25 | 41.03 | 39.41 | 38.05 |
| | Rare | 45.03 | 44.60 | 44.08 | 42.95 | 41.33 | 40.30 |
| | Non-Rare | 42.20 | 42.02 | 41.61 | 40.35 | 38.73 | 37.27 |
| **Box-Scale** | Full | 42.93 | 42.81 | 42.33 | 41.49 | 40.03 | 38.30 |
| | Rare | 45.03 | 44.89 | 43.92 | 42.92 | 41.03 | 39.10 |
| | Non-Rare | 42.20 | 42.09 | 41.78 | 40.96 | 39.63 | 37.97 |

Table 13: Robustness analysis under spatial perturbations. We report mAP for Full, Rare, and Non-Rare subsets across different perturbation strengths for both box shifting and box scaling.

the corresponding results are summarized in Table 13. Even under severe perturbation (e.g., $\delta = 1.0$), our method maintains over 38.0 mAP, which is comparable to the *undisturbed* performance of several state-of-the-art approaches (e.g., CMMP, ADA-CM, EZ-HOI). Under mild perturbation (e.g., $\delta = 0.2$), the performance degradation is negligible (less than 0.3 mAP). Redundant detections are not a concern in our setting, as they can be reliably handled using standard NMS suppression.

This robustness stems from our **VLM Linking Decoder**, which does not rely solely on geometric cues or ROI features. Each decoder layer attends to the global VLM representation, which remains stable even when box coordinates are perturbed, enabling reliable HOI reasoning. As illustrated in Fig. 11, small boxes perturbations do not affect the predictions of our LINK model.

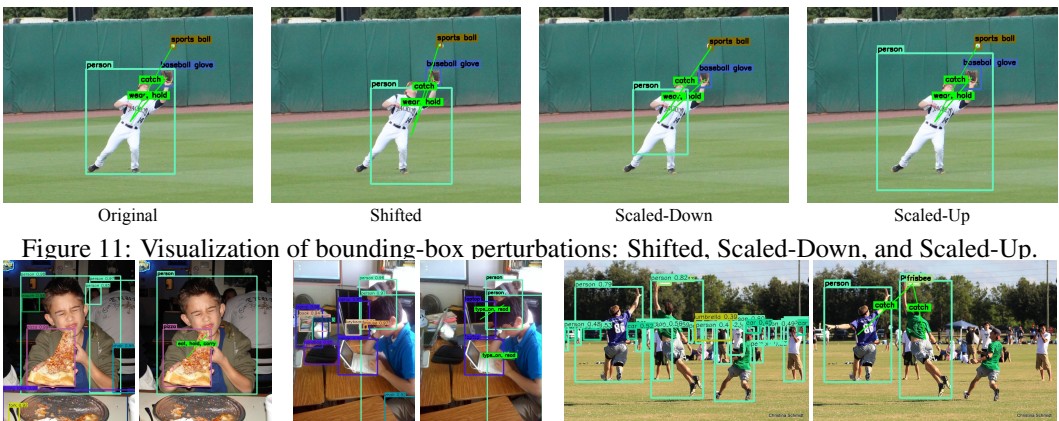

| Original | Shifted | Scaled-Down | Scaled-Up |

Figure 11: Visualization of bounding-box perturbations: Shifted, Scaled-Down, and Scaled-Up.

Figure 12: Robust HOI prediction under redundant detection boxes.

Moreover, the plug-and-play experiments with diverse open-vocabulary detectors (Table 12) further support this conclusion from a complementary perspective: although different OVDs produce bounding boxes that vary greatly in confidence distribution, density, and localization bias, our method remains fully training-free, plug-and-play, and consistently strong across all detectors.

## A.13 INSUFFICIENT SPATIAL UNDERSTANDING IN CLIP

(a) Layer-wise PCA Visualization of **CLIP-ViT-L/14@336px**

(b) Layer-wise PCA Visualization of **DINOv2-base**

Figure 13: **Layer-wise PCA visualization of patch embeddings. CLIP v.s DINOv2**

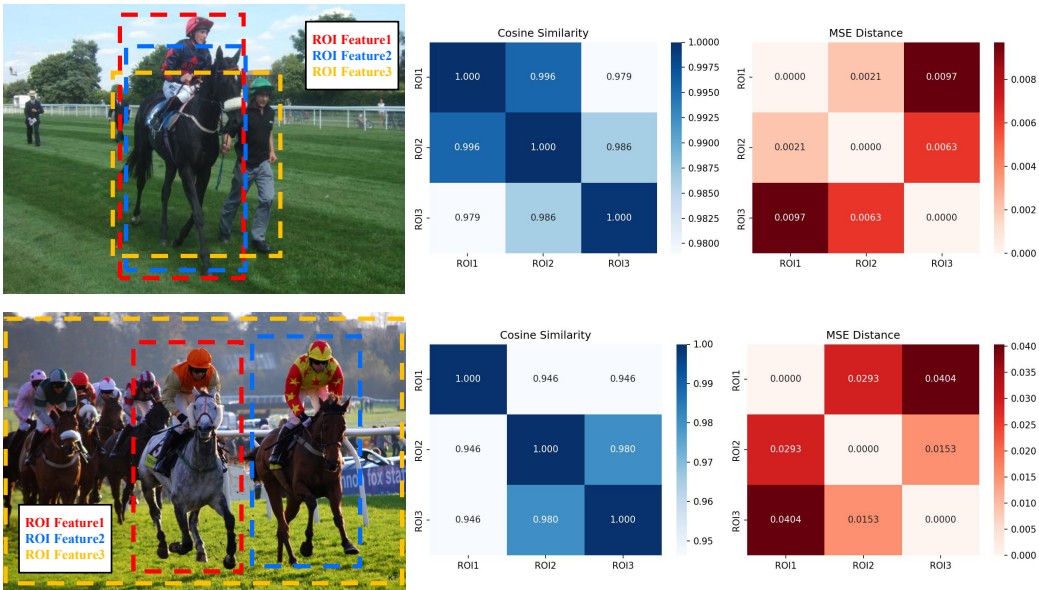

Figure 14: **CLIPs feature-map tokens exhibit insufficient spatial variability.**

