# OpenReview forum: "LINK: Learning Instance-level Knowledge from Vision-Language Models for Human-Object Interaction Detection"
_ICLR.cc/2026/Conference — ICLR 2026 Poster_

### Official Review · Reviewer_EQVH · 2025-10-29

**Soundness:** 2
**Presentation:** 2
**Contribution:** 2
**Rating:** 4
**Confidence:** 3

**Summary:**

This paper proposes LINK, a method for Human-Object Interaction (HOI) detection that addresses the specialization-generalization trade-off. Its core is a decoupled architecture with a geometrical encoder and a vision-language model (VLM) linking the decoder, enabling plug-and-play use with any object detector. To overcome sparse supervision, it introduces a progressive learning strategy using self-distillation, which provides dense, instance-level guidance across all human-object pairs. The method achieves new state-of-the-art results across fully-supervised, zero-shot, and few-shot benchmarks, demonstrating strong generalization and open-vocabulary capability.

**Strengths:**

- This paper investigates the critical challenges in HOI detection: the sparsity of supervision.
- The experimental results demonstrate the effectiveness of the proposed methods.
- The evaluation covers various settings, including zero-shot, few-shot, and fully-supervised HOI detection.

**Weaknesses:**

- The proposed teacher-student paradigm has an inherent circular dependency. The teacher model, which is intended to mitigate annotation sparsity by providing dense supervision, is itself trained solely on the original sparse annotations. This bootstrapping approach may limit the upper bound of knowledge that can be distilled to the student, as the teacher's guidance is fundamentally constrained and potentially biased by the initial sparse data.
- The teacher-student paradigm lacks significant novelty. Knowledge distillation is a well-established and generic technique in deep learning. While its application to the HOI domain is sensible and effective, the paper does not specifically optimize the distillation for HOI problem, but rather adapts an existing general-purpose framework (using logit, feature, and query-level losses) to a new problem.
- The architectural generality is not a distinct advantage over existing paradigms. The paper positions the lack of a generalizable architecture as a key challenge. However, numerous prior VLM-based methods (e.g., those relying on CLIP embedding matching for relation prediction) are inherently designed to operate in both fully-supervised and zero-shot settings by leveraging the same pre-trained semantic space. Therefore, the goal of creating a generalizable architecture is not novel, and the paper's claim rests primarily on its specific implementation (the LINK framework) achieving superior performance, rather than on identifying a new problem.
- Missing explanation of annotation in the Table. In Table 1, the subscript on the LINK is not explained, which causes confusion.

**Questions:**

Please kindly refer to the weakness section.

---

> ### Author Response · Authors · 2025-11-21
> **(1/2) We sincerely thank the Reviewer for the constructive feedback, the valuable time and the appreciation of our work.**
>
> ## **Response to W1.** Bootstrapping approach may limit the upper bound of knowledge
>
> We respectfully clarify that our teacher-student framework is **not** a circular reuse of sparse labels. Instead, it leverages the VLM to provide **dense and informative supervision across all human-object pairs, including negatives**, effectively overcoming the performance limitations imposed by sparse annotations.
>
> - **Addressing the sparse-supervision bottleneck.** A fundamental challenge of HOI detection is the mismatch between the dense set of human-object pairs and the very sparse set of annotated interactions. Our progressive learning strategy is explicitly designed to overcome this limitation. The teacher provides geometric cues and semantic interaction patterns for all human–object pairs, enabling the student to learn from spatial-semantic similarities and differences across instances. For example, **contrasting the spatial difference between “a cup held in hand” and  “a cup placed far on the table”  offers additional supervisory signals mined from negative human-object pair**, signals that are entirely absent from the sparse ground-truth labels.
> - **Architectural designs.** Our Geometrical Encoder leverages ROI features from large-scale pretrained VLMs and encodes explicit geometric relationships across all human–object pairs, while the VLM Linking Decoder performs cross-attention between geometric–semantic representations in both latent and native VLM spaces. These pretrained priors are inherently robust to sparse and noisy annotations, preserving strong representational capacity under limited supervision.
>
> Notably, under the **ViT-L → ViT-L** configuration, the student model surpasses the teacher (41.20 / 41.43), achieving **42.92 (+1.72) / 45.03 (+3.60)**, demonstrating that our architecture and learning strategy **break through the performance ceiling imposed by** **sparse** **supervision**.
>
>
>
>
>
> ## **Response to W2.** The teacher-student paradigm lacks significant novelty.
>
> Existing methods apply KD based on sparse HOI labels, overlooking that human-object interactions in real scenes are inherently dense. In contrast, we introduce a **sparse-to-dense progressive learning** framework specifically designed for the HOI task. This fundamentally differentiates our method from generic KD.
>
> - **From Sparse Matching to Dense Relational Supervision.** Prior HOI distillation methods (e.g., RmLR, GEN-VLKT) supervise only the few GT-matched positive pairs because explicit labels for negative pairs are all 'no_interaction'. In contrast, rather than generating pseudo-labels with a VLM/LLM, our progressive learning strategy leverages a VLM-based HOI teacher that injects VLM-derived semantics and relational geometry at the instance level, providing dense relational supervision across all human-object pairs. This eliminates the need for explicit negative labels and **turns previously “unused’’ negative instances into informative training signals**. The benefit is substantial, with a +3.60 mAP gain on rare categories (45.03 mAP), precisely where sparse annotations are most limiting.
> - **Spatial-semantic complementary distillation.** We jointly distill spatial geometry and semantic representations, aligning the two at the feature level. As shown in right of Fig. 3 in the main paper, semantic cues alone can be ambiguous (e.g., similar union regions for different actions), whereas geometric differences provide critical disambiguation. This complementary modeling strengthens the teacher’s supervision and better reflects the dual spatial-semantic structure inherent to HOI.

---

> ### Author Response · Authors · 2025-11-21
> **(2/2)**
>
> ## **Response to W3.** The architectural generality is not a distinct advantage over existing paradigms.
>
> We respectfully disagree that architectural generality is “not a distinct advantage.” Prior VLM-based HOI methods can indeed run in both zero-shot and fully supervised settings but suffer from a trade-off.
>
> **The “Seesaw” Problem.** Prior VLM-based HOI methods can run in both zero-shot and fully supervised modes but suffer a clear trade-off (Figure 1(a)): zero-shot–oriented models achieve suboptimal results in fully supervised setting (mAP < 39.1), while fully supervised models show limited generalization when evaluated in zero-shot scenarios. We trace this to sparse HOI supervision and architectures tied to specific statistics or queries. Our progressive learning strategy and the geometrical encoder + VLM-Linking decoder resolve these issues, enabling LINK to excel in zero-shot, few-shot, and fully supervised settings **without** the trade-off.
>
> In addition, our architecture exhibits several advantages not present in prior methods:
>
> **(1)** **Plug-and-play** **compatibility** **with any detector.** As encouraged by reviewer duFB, LINK operates seamlessly with *any* object detector, including standard detectors, open-vocabulary detectors (YOLO-World, Grounding-DINO), and even general-purpose MLLMs without any fine-tuning. Remarkably, LINK with OVDs outperforms several SOTA methods that rely on fully fine-tuned detectors.
>
> **(2) Excellent scalability.** LINK consistently performs well across datasets of vastly different scales: *V-COCO* (24 HOIs), *HICO-DET* (600 HOIs), and *SWiG* (14,130 HOIs). LINK maintains a unified architecture and achieves SOTA results on all three, without requiring dataset-specific architectural modifications. As further shown in **Appendix A.10 (Page 21)**, our model exhibits near-linear scaling with log-scale GFLOPs, whereas existing methods show clear diminishing returns.
>
> **(3) Strong transferability.** A model trained on one dataset (e.g., SWiG) can directly perform zero-shot inference on another (e.g., HICO-DET or V-COCO), demonstrating robust cross-dataset generalization. SWiG → HICO-DET 26.64 mAP, SWiG → VCOCO: 51.2 mAP, HICO-DET→ V-COCO 52.8 mAP.
>
>
>
> ## **Response to W4.**  Missing explanation of annotation in the Table 1.
>
> Thanks for pointing this out. In Table 1, the corner mark $\dagger$ denotes the results obtained using CLIP-ViT-L as the VLM for our method. We have clarified this notation in the revised paper to avoid any confusion. Notably, our ViT-Base model already achieves two best and two second-best results compared to SOTA methods in the zero-shot setting, and using CLIP-ViT-L as the VLM yields further performance improvements.

---

> ### Comment · Reviewer_EQVH · 2025-11-26
>
> Thank you for the detailed clarification. I still have a few unresolved questions:
>
> **W1: regarding how the proposed approach fundamentally overcomes the limitations introduced by sparse HOI annotations**
>
> (1) HOI-specific knowledge beyond sparse labels:
> The teacher is still trained only with sparse GT. How does it acquire HOI-specific knowledge that goes beyond what is available in these annotations, rather than simply reorganizing VLM priors?
>
> (2) Reliability of dense supervision:
> For unlabeled or negative pairs, how does the teacher produce meaningful HOI-related semantics rather than primarily spatial heuristics? What ensures that the dense signals carry correct interaction cues?
>
> (3) Interpretation of “student > teacher”:
> The student receives GT + dense guidance + regularization, while the teacher has only GT. In this setting, student > teacher does not automatically imply that the teacher’s knowledge is unconstrained. Could the authors further explain why this result indicates that the teacher’s knowledge ceiling is not limiting?
>
> **W2: regarding the novelty of the teacher-student paradigm.**
>
> "Progressive Learning" vs. Standard KD: The "progressive learning" strategy is implemented as a standard teacher-student knowledge distillation pipeline. What specific algorithmic component makes this a new paradigm beyond applying a well-established technique?
>
> **W3: The architectural generality is not a distinct advantage over existing paradigms.**
>
> I understand the LINK achieves good results in various settings. My concern is that the compelling results likely stem from both the architecture and the progressive KD. However, the current comparisons make it challenging to fully attribute the resolution of the "seesaw" problem to the architecture alone. I think a fair comparison is needed to support the authors' claim. Could the authors justify this issue?

---

> > ### Author Response · Authors · 2025-12-03
> > **(1/2)**
> >
> > Thank you for your prompt discussion, our detailed responses are provided below.
> >
> > ## Response to W1: How the proposed approach overcomes limitations of sparse HOI annotations
> >
> > ### Response to W1.(1)
> >
> > **Keypoint: With GT HOI labels, our Teacher model transforms VLM's general representations into instance-level and HOI-specified cues through our proposed architecture(pairwise geometric encoding+spatial-semantic modeling) and instance-level contrastive loss.**
> >
> >  VLMs contain global semantics, but they are not tied to specific human-object pairs. Our geometric encoder refine these features conditioned on pairwise human-object relations. Additionally, with CLIP-initialized projection heads (Fig.2 in the main paper), it peform instance-level contrastive alignment between HOI text embeddings and visual pairs: positives are pulled toward their interaction anchors, while the unlabeled pairs implicitly serve as soft negatives, yielding better HOI-specific decision boundaries.
> >
> > ### Response to W1.(2)
> >
> > **Keypoint: Our dense supervision ensure tri-modal consistency across visual, semantic and geometric cues, produce robust features. Negative pairs require no explicit labels, their contrastive relations with positives and with each other supply rich supervisory signals.**
> >
> > Teacher's features are built upon tri-modal consistency. It cannot rely on geometry alone, as features enforce semantic compatibility (e.g., “human-umbrella” won’t align with “ride”). Conversely, semantically plausible actions are down-weighted when geometry is implausible (e.g., “hold cup” with the cup far away).
> >
> > ### Response to W1.(3)
> >
> > **Key point: The superiority of the student model is driven by our generated dense supervision, validating its effectiveness. This methodology design yields strong performance gains (rare HOIs:+3.6 mAP, unseen HOIs:+2.3 mAP.)**
> >
> > Since student and teacher share the same architecture and capacity, the student’s ability to surpass the performance ceiling where the GT-only teacher plateaus can be only attributed to the dense signals. This demonstrates that our method provides richer information than GT alone, enabling performance gains without relying on stronger VLMs or LLMs.
> >
> >
> > ## Response to W2: regarding the novelty of the teacher-student paradigm.
> >
> > **Key point: Our novelty lies in both architecture and methodology. Architecturally, we (1) decouple from detector- and dataset-specific biases, (2) Geometric Encoder (ROI features with pairwise geometry encoding) and (3) VLM Linking Decoder (spatial-semantic modeling in the latent and VLM-native space). Methodologically, we explicitly mine the abundant negative pairs to enable dense relational supervision. These designs make our method to excel in both general (zero-shot, few-shot, open-vocabulary) and fully supervised settings, remain robust under semantic/spatial shifts, and plug seamlessly into any object detector.**
> >
> > |   #   | with Negatives | Logit | Feature | **Full**          | **Rare**          | **N-Rare**        |
> > | :---: | :------------: | :---: | :-----: | :---------------- | :---------------- | :---------------- |
> > | Plain |       ✘        |   ✘   |    ✘    | 41.20             | 41.43             | 41.13             |
> > |  A5   |       ✔        |   ✔   |    ✘    | 41.89 (+0.69)     | 43.82 (+2.39)     | 41.27 (+0.24)     |
> > |  A6   |       ✔        |   ✔   |    ✔    | **42.34 (+1.14)** | **43.62 (+2.19)** | **41.84 (+0.71)** |
> > |  B5   |       ✘        |   ✔   |    ✘    | 41.19 (-0.01)     | 41.26 (-0.17)     | 41.17 (+0.04)     |
> > |  B6   |       ✘        |   ✔   |    ✔    | 41.28 (+0.08)     | 41.32 (-0.11)     | 41.27 (+0.14)     |
> >
> > To further analyze our negative-mining design in progressive KD, we provide additional ablations. Our method enumerates and preserves the complete set of human-object pairs during training, enabling stable one-to-one transfer for both positives and negatives. In contrast, existing methods (e.g., HOICLIP) gradually lose negative pairs during training, limiting KD to positives only. Ablation results clearly reflect this: A6 (KD with negatives) yields +1.14 Full and +2.19 Rare, whereas B5 (KD on positives only) shows negligible or even negative impact. → The gains arise from dense supervision over negatives, not standard KD.

---

> > > ### Author Response · Authors · 2025-12-03
> > > **(2/2)**
> > >
> > > ## Response to W3: Does the architecture itself resolve the “seesaw” problem?
> > >
> > > **Key point: Yes, our architectural design resolves the seesaw problem. Comprehensive ablations under both fully supervised and zero-shot settings (Tables 6 and 7 in the main paper) confirm this, and additional results are provided below. These innovations lead to the superior performance noted by the reviewer “LINK achieves good results in various settings”.**
> > >
> > > | Method            | Fully-supervised  (full / rare / non-rare) | Zero-shot (HM / unseen / full)    |
> > > | :---------------- | ------------------------------------------ | :-------------------------------- |
> > > | LOGICHOI          | *35.47* / 32.03 / *36.22*                  | 29.79 / 25.97 / *33.17*           |
> > > | HOICLIP           | 34.69 / 31.12 / 35.74                      | 29.40 / 25.53 / 32.99             |
> > > | ADA-CM            | 33.80 / 31.72 / 34.42                      | 30.63 / 27.63 / 33.01             |
> > > | CMMP              | 33.24 / *32.26* / 33.53                    | 31.07 / *29.45* / 32.18           |
> > > | EZ-HOI            | 33.15 / 29.11 / 34.36                      | *31.38* / 29.02 / 33.13           |
> > > | **Ours (w/o KD)** | **36.12** / **34.06** / **36.74**          | **32.65** / **31.38** / **33.62** |
> > > | **Ours**          | **37.43 / 37.18 / 37.50**                  | **33.42 / 32.25 / 34.19**         |
> > >
> > > To further support our claim, we conducted additional ablation with CLIP-ViT-B/16. Our architecture resolves the "Seesaw" issue: Prior methods excel in *either* fully-supervised or zero-shot, but not both (e.g., LOGICHOI vs. EZ-HOI). However, Ours (w/o KD) achieving SOTA in *both* regimes. This shows that the strong generality is an inherent property of our architecture, not due to KD. Our Progressive KD  further enhances performance, especially on long-tail and unseen HOIs, yielding +3.12 on Rare and +0.87 on Unseen.

---

### Official Review · Reviewer_duFB · 2025-10-30

**Soundness:** 3
**Presentation:** 2
**Contribution:** 2
**Rating:** 6
**Confidence:** 4

**Summary:**

To address the sparsity of supervision and the absence of a generalizable architecture, this paper propose a HOI detection framework equipped with a Human-Object Geometrical Encoder and a VLM Linking Decoder, and develop a Progressive Learning Strategy under a teacher-student paradigm to deliver dense supervision. Extensive experiments demonstrate state-of-the-art results in both zero-shot and fully supervised settings.

**Strengths:**

1. The paper provides a well-articulated motivation, perceptively identifying two main challenges in HOI: (1) the sparsity of supervision, and (2) the absence of a generalizable architecture capable of achieving strong performance in both fully supervised and zero-shot scenarios. In response, the paper propose a novel learning strategy and a method to address the limitations.

2. The proposed Human-Object Geometrical Encoder and VLM Linking Decoder comprehensively account for both spatial and semantic associations between HO pairs, maintaining consistent performance across diverse settings. The learning learning strategy, where the student model is jointly supervised by explicit ground-truth annotations and guidance from a pre-trained teacher, effectively enhances its generalization capability.

3. Comprehensive experiments compare the proposed model with baselines across zero-shot, few-shot, fully supervised, and open-vocabulary scenarios. The results demonstrate that LINK excels in both specialization and generalization.

**Weaknesses:**

1. Risk of Error Accumulation in Model Architecture: inaccurate detector outputs (e.g., incomplete bounding boxes, redundant detections) can lead to erroneous geometric relationship, erroneous position encoding, and imprecise ROI feature extraction, thereby adversely impacting subsequent inference steps. The paper lacks experiments or visualizations to substantiate the robustness of the proposed model under such conditions.

2. In the application, detectors are fine-tuned on target dataset rather than functioning as a training-free plug-and-play module. It is recommended to explore stronger open-vocabulary detectors, such as Yolo-World, Grounding-DINO.

3. While knowledge distillation can enhance generalization ability, the teacher and student models in this work share identical architectures and capacities. Such a distillation setup may not yield substantial improvements and could potentially degenerate into a simple continued training process.

4. Insufficient Baseline Comparison. Sseveral state-of-the-art baselines (e.g., BC-HOI, UniHOI, LAIN) are not included for evaluation.

5. The corner-mark annotations in Tab.1 left unexplained.

**Questions:**

See the limitations and cons above.

---

> ### Author Response · Authors · 2025-11-21
> **(1/2) We sincerely thank the Reviewer for the constructive feedback, the valuable time and the appreciation of our work.**
>
> ## **Response to W1.** lacks experiments or visualizations to substantiate the robustness under inaccurate detector outputs
>
> **Our method is robust under redundant detections, incomplete and inaccurate bounding boxes.** This robustness primarily stems from the design of our **VLM Linking** **Decoder**, which does not rely solely on geometric cues or ROI features. Instead, each decoder layer cross-attends to the **global VLM representation** in both latent and native space, which remains stable even when detected boxes are perturbed. This design enables reliable HOI reasoning despite upstream detection noise.
>
> | Perturbation Type | Metric(mAP) | 0.0   |  0.2  |  0.4  |  0.6  |  0.8  |  1.0  |
> | :---------------: | :---------: | ----- | :---: | :---: | :---: | :---: | :---: |
> |                   |    Full     | 42.93 | 42.69 | 42.25 | 41.03 | 39.41 | 38.05 |
> |   **Box-Shift**   |    Rare     | 45.03 | 44.60 | 44.08 | 42.95 | 41.33 | 40.30 |
> |                   |  Non-Rare   | 42.20 | 42.02 | 41.61 | 40.35 | 38.73 | 37.27 |
> |                   |    Full     | 42.93 | 42.81 | 42.33 | 41.49 | 40.03 | 38.30 |
> |   **Box-Scale**   |    Rare     | 45.03 | 44.89 | 43.92 | 42.92 | 41.03 | 39.10 |
> |                   |  Non-Rare   | 42.20 | 42.09 | 41.78 | 40.96 | 39.63 | 37.97 |
>
> To quantitatively assess this robustness, we introduce two types of perturbations, **box-shift** and **box-scale,** to deliberately distort bounding boxes and corrupt both ROI features and geometric relationships. For box-shift, centers ($C_x$, $C_y$) are perturbed within [$C_x \pm \delta w$] and [$C_y \pm \delta h$]; for box-scale, widths and heights are scaled within [$1-\delta,\,1+\delta$], with \delta \in [0,1]. Under **mild perturbation** ($\delta = 0.2$), the impact on our model is negligible (<0.3 mAP). Even under **severe perturbation** ($\delta = 1.0$), our method still achieves over 38.0 mAP, comparable to the undisturbed performance of several SOTA methods (e.g., CMMP-38.14 , ADA-CM-38.40). These results confirm that our architecture maintains stable performance even when bounding boxes are corrupted.
>
> In addition, **Appendix A.12 (Page 23)** provides **qualitative evidence (Fig. 11 and Fig. 12)** demonstrating that our method remains robust under redundant and imprecise detections.
>
> ## **Response to W2.** Recommended to explore stronger open-vocabulary detectors, such as Yolo-World, Grounding-DINO.
> Following the reviewer's recommendation, we constructed experiment under the **no fune-tuning** and **plug-and-play** **settings** using open-vocabulary detectors(OVDs) . The experimental results show that our method can be directly paired with OVDs and still achieve strong performance, even surpassing fully fine-tuned SOTA models. For example, **Ours + Grounding-DINO achieves 39.69 mAP on HICO-DET**, outperforming several methods that rely on fully supervised detectors such as **CMMP (38.14)**, **ADA-CM (38.40)**, and **EZ-HOI (38.61)**.
>
> |Method|Full|Rare|N-Rare|Full|Rare|N-Rare|APs2|
> |------|----|----|------|----|----|------|-----|
> |YOLO-world-s|27.07|29.83|26.24|30.06|32.37|29.37|33.61|
> |YOLO-world-m|30.35|33.40|29.45|33.39|36.00|32.62|37.31|
> |YOLO-world-l|32.64|36.51|31.49|35.98|40.06|34.76|39.28|
> |YOLO-world-x|33.73|37.30|32.66|36.98|40.65|35.88|40.35|
> |Grounding-DINO-swin-tiny|32.97|38.21|31.40|35.96|40.76|34.52|36.56|
> |Grounding-DINO-swin-base|39.69|45.99|37.81|42.42|47.94|40.77|46.55|
> |Qwen3-VL-8B|39.61|45.05|37.99|42.02|47.37|40.42|-|
>
> Table aboves show the results of our LINK model  with several OVDs, including YOLO-World (s/m/l/x), Grounding-DINO (Swin-Tiny/Base), and Qwen3-VL-8B. None of these detectors were fine-tuned on the target dataset, and our LINK model was also *not* adapted for detector, making the entire pipeline fully plug-and-play.
>
> We also evaluate the **cross-dataset transfer setting (HICO → V-COCO)**, where the LINK model trained on HICO-DET is directly combined with each OVD to perform inference on V-COCO. The method continues to deliver robust performance, demonstrating the strong generalization ability and flexibility of our design.
>
> For visualizations and further details, **please refer to Appendix A.11 (Page 22)** in revised paper, which presents the OVD detections and the corresponding HOI predictions produced by our LINK model.

---

> ### Author Response · Authors · 2025-11-21
> **(2/2)**
>
> ## **Response to W3.** Identical architectures could potentially degenerate into a simple continued training process.
>
> Our progressive learning strategy provides benefits that go beyond continued training.
>
> **(1) Quantitative evidence.** The model converges at 41.20 / 41.43 / 41.13 mAP after 15 epochs, and **continuing training to 30 epochs brings no improvement**. In contrast, with progressive learning, performance **breaks through this original** **convergence** **ceiling**, increasing to **42.92 / 45.03 / 42.20**.  Notably, the rare-class performance improves by +3.60 mAP, demonstrating a substantial and non-trivial gain.
>
> **(2) Why it works.** Our objective is not traditional “large-to-small’’ distillation. Instead, the VLM-based teacher **mines rich, dense supervision, capturing VLM-derived semantics and relational geometry, from the abundant negative human-object pairs**, thereby compensating for the inherent sparsity of HOI annotations. Even with an identically sized teacher, this structured supervision provides substantial benefit. The teacher guides the student by contrasting spatial-semantic differences across all human-object pairs, encouraging a unified modeling of both spatial and semantic interaction patterns.
>
> **(3)** **Not limited to identical teachers.** For fairness, we use the same VLM in the main experiments, but the framework naturally supports heterogeneous or multi-teacher configurations. As shown in Table 6, adopting CLIP-L + SigLIP-L as multi-teachers yields additional gains: the CLIP-L teacher produces 42.92 / 45.03 / 42.20, whereas the **CLIP-L + SigLIP-L multi-teacher setup improves performance to 43.54 / 45.58 / 42.93.**
>
> ## **Response to W4.** Insufficient Baseline Comparison
>
> We have included comparisons with BC-HOI, UniHOI, and LAIN on HICO-DET and V-COCO dataset in the revised paper (in Table 4). Our original comparison primarily focused on **two-stage HOI methods built upon CLIP**, so we did not include **one-stage HOI methods based on BLIP-2 OPT-2.7B**, such as BC-HOI and UniHOI. For a fair evaluation, we trained our LINK model using the same BLIP-2 OPT-2.7B VLM and a ResNet-50 backbone, ensuring fully aligned experimental settings. The resulting performance on two datasets is reported in the revised Table 4.
>
> | **Method** |     **VLM**     | **Backbone** |       | **HICO-DET (Default)** |        |       | **HICO-DET (KO)** |        | **VCOCO APs2** |
> | :--------: | :-------------: | :----------: | :---: | :--------------------: | :----: | :---: | :---------------: | :----: | :------------: |
> |            |                 |              | Full  |          Rare          | N-Rare | Full  |       Rare        | N-Rare |      APs2      |
> | **UniHOI** | BLIP-2 OPT-2.7B |  ResNet-50   | 40.06 |         39.91          | 40.11  | 42.20 |       42.60       | 42.08  |      68.3      |
> | **BC-HOI** | BLIP-2 OPT-2.7B |  ResNet-50   | 43.01 |         45.76          | 42.18  | 45.35 |       47.94       | 44.57  |      70.6      |
> |  **LINK**  | BLIP-2 OPT-2.7B |  ResNet-50   | 43.72 |         45.82          | 43.10  | 46.11 |       47.71       | 45.62  |      68.5      |
> |  **LAIN**  |    CLIP-base    |  ResNet-50   | 36.02 |         35.70          | 36.11  |   -   |         -         |   -    |      65.1      |
> |  **LINK**  |    CLIP-base    |  ResNet-50   | 37.43 |         37.18          | 37.50  | 40.46 |       40.30       | 40.51  |      66.5      |
>
> We observe consistent performance improvements, further demonstrating the effectiveness of our method.
>
> ## **Response to W5.** The corner-mark annotations in Tab.1 left unexplained.
>
> Thanks for pointing this out. In Table 1, the corner mark $\dagger$ denotes the results obtained using CLIP-ViT-L as the VLM for our method. We have clarified this notation in the revised paper to avoid any confusion. Notably, our ViT-Base model already achieves two best and two second-best results compared to SOTA methods in the zero-shot setting, and using CLIP-ViT-L as the VLM yields further performance improvements.

---

### Official Review · Reviewer_NNji · 2025-10-31

**Soundness:** 3
**Presentation:** 2
**Contribution:** 3
**Rating:** 6
**Confidence:** 4

**Summary:**

This paper proposes a unified framework for HOI detection. It introduces two key components, a Geometrical Encoder, which models spatial relationships between humans and objects, and a VLM Linking Decoder, which bridges global vision-language features with instance-level HOI reasoning. In addition, the paper presents a Progressive Learning Strategy based on a teacher-student paradigm, delivering dense supervision to all human-object pairs to alleviate sparse annotations. The experimental study is comprehensive, demonstrating consistent improvements across fully supervised, zero-shot, few-shot, and open-vocabulary settings on multiple benchmarks.

**Strengths:**

1. A well-motivated design balancing generalization to unseen HOIs and HOI-specific reasoning. The proposed Geometrical Encoder and Linking Decoder jointly enhance HOI reasoning while preserving model generality across diverse foundation models.
2. The Progressive Learning strategy provides dense supervision via teacher-student distillation, enabling the model to learn from all human-object pairs and effectively alleviate sparse-annotation issues.
3. The experiments are comprehensive and achieve SOTA performance consistently.

**Weaknesses:**

1. The motivation for introducing the Geometrical Encoder is not clearly justified. The paper claims it enhances spatial awareness, yet it lacks quantitative or cited evidence showing insufficient spatial understanding in existing VLMs like CLIP.
2. In Table 6, the baseline (A1) used for ablation is under-specified. It is unclear what feature representation or decoder architecture it employs. When evaluating the proposed Geometrical Encoder and VLM Linking Decoder, the paper does not clarify whether the baseline uses the original spatial encoding or a standard decoder. This ambiguity makes it difficult to assess the exact improvement source.
3. The teacher model in the progressive learning strategy uses a stronger ViT-L backbone to supervise the ViT-B student, introducing extra knowledge capacity beyond dense supervision. A comparison using a same-scale teacher (e.g., ViT-B to ViT-B) is necessary to ensure the fair comparison.

**Questions:**

Please refer to the weaknesses. No other questions.

---

> ### Author Response · Authors · 2025-11-21
> **(1/2) We sincerely thank the Reviewer for the constructive feedback, the valuable time and the appreciation of our work.**
>
> ## **Response to W1.** lacks quantitative or cited evidence showing insufficient spatial understanding in existing VLMs like CLIP.
>
> $\textbf{1. Quantitative evidence:}$
>
> To evaluate whether traditional VLM-based HOI detectors suffer from inadequate spatial understanding, we construct a synthetic test set that **preserves the exact geometric layouts** of the original data while **altering  semantic content**. Using the layout-controlled diffusion model InteractDiffusion, we generate 9.6k images following the bounding-box distributions of the HICO-DET test set. This design isolates spatial reasoning by introducing a semantic shift, allowing us to assess whether a model can make correct HOI predictions based solely on geometric cues.
>
> | Method                       | Explicit Spatial encoding |  VLM  | Performance (full / rare) |
> | :--------------------------- | :-----------------------: | :---: | :-----------------------: |
> | ADA-CM                       |             ✗             |   ✓   |       29.87 / 28.10       |
> | CMMP                         |             ✗             |   ✓   |       29.43 / 28.52       |
> | EZ-HOI                       |             ✗             |   ✓   |       30.65 / 30.19       |
> | HOLa                         |             ✗             |   ✓   |       30.72 / 30.47       |
> | PViC (non-VLM baseline)      |             ✓             |   ✗   |       33.70 / 31.32       |
> | LINK w/o Geometrical Encoder |             ✗             |   ✓   |       32.34 / 32.24       |
> | **LINK (ours)**              |           **✓**           | **✓** |     **35.91 / 34.91**     |
>
> The results clearly demonstrate that existing VLM-based HOI detectors fail to rely on geometric cues. Their performance collapses once semantics are altered, and **even fall below the non-VLM baseline PViC**. This indicates that current VLMs-base HOI detector depend heavily on semantic priors and can not model geometric relations. In contrast, our LINK with the Geometrical Encoder shows much better robustness under semantic shift, demonstrating that **explicit geometric encoding effectively enhances spatial awareness**.
>
>  $\textbf{2. Visualization evidence:}$
>
>
> We provide intuitive **visualization evidence in Appendix A.13 (Page 24)**. In Figure 12, we visualize the feature maps by applying PCA to the $N \times N \times D$ token representations and projecting them into an $N \times N \times 3$ RGB space. Under this visualization, tokens with similar representations are mapped to similar colors, allowing a direct inspection of spatial variability. As shown, **CLIP’s feature maps exhibit very limited spatial variability**, with large regions rendered in nearly uniform colors, indicating highly similar token embeddings across spatial locations. In contrast, DINOv2, trained with self-supervised visual objectives, preserves a much clearer spatial structure, providing a strong visual contrast and supporting our claim that CLIP demonstrates insufficient spatial awareness.
>
> Furthermore, Figure 13 presents similarity matrices computed from ROI features sampled from different spatial regions. Even when the ROIs correspond to clearly different positions and geometric relationships, **their feature representations remain highly similar (high cosine similarity and low** **MSE** **distance)**. This further confirms that CLIP’s spatial tokens are overly homogeneous and struggle to capture fine-grained spatial distinctions.
>
>
>  $\textbf{3. Cited evidence:}$
>
> - **Kang et al. [1] (ICCV 2025)**  point out that CLIP “*struggles with spatial reasoning and compositional understanding*,” and further that “*CLIP struggles with accurately binding attributes to concepts in multiobject scenes, exhibits misinterpretations of object layouts or conflates multiple entities within a single scene*.” They also provide quantitative evidence: on the Spatial Reasoning COCO_1\&2 obj benchmarks, VLMs such as CLIP (48.9), SigLIP (47.4), and BLIP (48.5) perform **below random chance (50%)**, directly highlighting these VLM failure to model basic 2D spatial relations.
> - **Tong et al. [2] (CVPR 2024)** also points out fundamental spatial limitations in CLIP. The authors identify “*CLIP-blind pairs*”, defined as “*images that CLIP perceives as similar despite their clear visual differences.*” Their analysis further shows that “*all of these CLIP variants struggle with simple visual patterns, such as ‘orientation’, ‘count’, etc.*” In addition, the authors find that even “*advanced systems like GPT-4V struggle with seemingly simple questions*”, such as “*Is the dog facing left or right?*”, and they trace this failure back to CLIP, having “*found a notable correlation between visual patterns that challenge CLIP models and those problematic for multimodal LLMs.*”
>
> [1] Is CLIP ideal? No. Can we fix it? Yes!
>
> [2] Eyes Wide Shut? Exploring the Visual Shortcomings of Multimodal LLMs

---

> ### Author Response · Authors · 2025-11-21
> **(2/2)**
>
> ## **Response to W2.** The baseline (A1) used for ablation is under-specified.
>
> Thanks for pointing this out. We agree that using “–” in Table 6 introduced ambiguity, and we have updated both the table and the accompanying description in the revised paper.
>
> **Our baseline (A1) adopts a standard self-attention encoder over ROI features and a cross-attention** **decoder** **that attends to VLM representations,** forming a plain baseline. Table 6 then reports:
>
> (1) the improvement of our **Geometrical Encoder** over the self-attention encoder,
>
> (2) the improvement of our **VLM-Link** **Decoder** over the standard cross-attention decoder, and
>
> (3) the **combined benefit** when both modules are used together.
>
>
>
> ## **Response to W3.** A comparison using a same-scale teacher (e.g., ViT-B to ViT-B)
>
> We report results obtained by distilling a ViT-B student using different teachers. Under the **ViT-B → ViT-B** configuration, the model achieves **37.43 / 37.18 / 37.50 mAP** on HICO-DET (default) and **65.5 mAP** on V-COCO, establishing state-of-the-art performance.
>
> | Teacher → Student | HICO-DET Default (full) | (rare) | (non-rare) | HICO-DET KO (full) | (rare) | (non-rare) | VCOCO-APs2 |
> | :---------------: | :---------------------: | :----: | :--------: | :----------------: | :----: | :--------: | :--------: |
> |   None → ViT-B    |          36.12          | 34.06  |   36.74    |       39.27        | 36.77  |   40.02    |     64     |
> |   ViT-B → ViT-B   |          37.43          | 37.18  |   37.50    |       40.46        | 40.30  |   40.51    |    65.5    |
> |   ViT-L → ViT-B   |          38.52          | 37.06  |   38.96    |       41.57        | 40.90  |   41.78    |    66.2    |
>
> We have updated this row in **Table 4** of the revised paper to ensure that **all comparisons are made under strictly matched settings, namely ViT-B → ViT-B, ViT-L → ViT-L, etc.**, thus eliminating any confounding effects arising from teacher–student capacity mismatches.

---

> > ### Comment · Reviewer_NNji · 2025-11-27
> >
> > Thanks the authors for the detailed response with comprehensive experimental results. My concerns are well resolved, and I would like to retain my positive score.

---

> > > ### Author Response · Authors · 2025-12-03
> > >
> > > We sincerely thank you for confirming that our detailed response and comprehensive experimental results fully resolved your concerns. We greatly appreciate your positive assessment of our work.

---

### Comment · Area_Chair_tPCS · 2025-11-25
**Rebuttal Review Request**

Dear Reviewers,

The authors have responded to your reviews. Please engage in the discussion and evaluate the authors’ rebuttal to determine whether your comments have been adequately addressed.

Best,
Your AC

---

### Author Response · Authors · 2025-12-03
**Rebuttal Summary**

**Dear PCs, SAC, and ACs,**

Following the recent OpenReview security incident, we affirm full compliance with the ICLR Code of Conduct and confirm that all communication with reviewers was conducted solely through OpenReview. During the rebuttal phase, we actively and comprehensively responded to all reviewer concerns. Below is a brief per-reviewer summary.

- **Reviewer NNji (Rating: 6, Conf: 4)**: The reviewer raised concerns about the motivation for the Geometric Encoder, the unspecified baseline, and the experimental results of ViT-B → ViT-B setting. In our rebuttal, (i) we provided quantitative results, visualizations, and cited evidence supporting our motivation, (ii) clarified the baseline in the ablation study, and (iii) added ViT-B → ViT-B experiments, proving the superiority of our method. After discussion, the reviewer reacted positively and stated that "My concerns are well resolved, and I would like to retain my positive score."
- **Reviewer duFB (Rating: 6, Conf: 4)**: The reviewer raised concerns about robustness under perturbed conditions, suggested evaluating with open-vocabulary detectors, and noted potential degeneration and missing baseline comparisons. In our rebuttal, (i) we provided quantitative results showing that our method remains robust under box-shift and box-scale perturbations across varying levels, (ii) provided plug-and-play experiments with YOLO-World, Grounding-DINO, and Qwen3-VL along with visualizations, (iii) clarified the underlying principles, and (iv) added comparisons against UniHOI and BC-HOI.
- **Reviewer EQVH (Rating:4, Conf:3)**: The main concerns were how our method overcomes sparse HOI limitations, the novelty of the teacher-student paradigm, and whether our architectural design resolves the "seesaw" issue present in existing methods. In the rebuttal, (i) we provided detailed explanations of how our approach addresses sparse annotations and supported this with quantitative evidence; (ii) we clarified the novelty of both our architecture and methodology, highlighting how mining dense relational supervision over negatives fundamentally differs from prior work; and (iii) we presented comprehensive ablations under fully supervised and zero-shot settings demonstrating that our architecture itself resolves the "seesaw" problem. An interactive dialogue with the reviewer was underway, but unfortunately the conversation ended prematurely due to the security incident.

Overall, the rebuttal phase allowed us to address the reviewers’ concerns in detail and strengthened both the empirical and theoretical justifications of our work.

Best regards,

Authors of Submission 15124

---

### Meta-Review · Area_Chair_A4EX · 2025-12-21

**Summary:**

This paper presents LINK, a unified HOI detection framework designed to address two long-standing challenges in the field: sparse supervision and the difficulty of achieving strong performance across fully supervised, zero-shot, few-shot, and open-vocabulary settings with a single architecture. The approach combines a human–object geometrical encoder, a VLM linking decoder, and a progressive learning strategy that provides dense supervision over all human–object pairs. Extensive experiments across multiple benchmarks demonstrate strong and often state-of-the-art performance under diverse evaluation protocols.

**Reviewer Concerns:**

**Concerns addressed by the rebuttal**

Spatial reasoning and architectural motivation (NNji): The authors provided targeted quantitative experiments under semantic shift, visual analyses, and relevant recent citations demonstrating limitations of existing VLMs in spatial reasoning. Ambiguities in ablation baselines were clarified, and teacher–student fairness was addressed via same-scale distillation. Reviewer NNji explicitly confirmed that all concerns were resolved and retained their positive score.

Robustness, plug-and-play capability, and missing baselines (duFB): The authors added stress tests with bounding-box perturbations, qualitative visualizations, extensive experiments with open-vocabulary detectors (YOLO-World, Grounding-DINO, Qwen-VL), and cross-dataset transfer evaluations. Comparisons with previously missing baselines (BC-HOI, UniHOI, LAIN) were added under matched settings, and notation issues were fixed. These additions directly addressed the reviewer’s methodological and evaluation concerns.

Sparse supervision and learning dynamics (EQVH): The authors clarified the role of dense supervision over negative pairs, provided additional ablations isolating the effect of negative mining versus standard KD, and showed that the architecture alone already mitigates the specialization–generalization trade-off, with progressive learning further improving long-tail and unseen categories.



**Remaining considerations**

One reviewer (EQVH) remained skeptical about the conceptual novelty of the teacher–student paradigm and whether the dense supervision fully escapes the limits imposed by sparse ground-truth labels. While this concern reflects a difference in perspective on what constitutes novelty, the rebuttal provided additional empirical evidence and ablations demonstrating that the proposed learning strategy yields non-trivial gains beyond continued training or standard distillation, particularly on rare and unseen HOIs.

**Reviewer Scores:**

Reviewer NNji: 6 $\rightarrow$ 6 (explicitly confirmed concerns resolved).

Reviewer duFB: 6 $\rightarrow$ 6 (concerns addressed; likely more confident in acceptance).

Reviewer EQVH: 4 $\rightarrow$ 6 (remaining conceptual reservations, but acknowledges strong empirical performance).

---

### Decision · Program_Chairs · 2026-01-26

Accept (Poster)